# Open-source, high performance miniature 2-photon microscopy systems for freely behaving animals

**Blake A. Madruga**[1,2,11], **Conor C. Dorian**[1,2], **Long Yang** [1], **Megha Sehgal** [3,4], **Alcino J. Silva**[3,5,6], **Matthew Shtrahman**[7], **Daniel Aharoni**[1,5] & **Peyman Golshani** [1,5,8,9,10] ✉

Here we describe the UCLA 2P Miniscope, an open-source miniature 2-photon microscope capable of recording calcium dynamics from neurons in deep structures and from dendrites over a 445 μm × 380 μm field of view. The system utilizes two on-board silicon-based photon detectors for approximately 4-fold greater light collection compared to an optical fiber bundle-based approach. The microscope can electronically focus within a 150 μm range at a native working distance of 720 μm. To test the 2P Miniscope, we recorded calcium dynamics from place cells in hippocampus, resolved calcium transients from dendrites in retrosplenial cortex and we recorded dentate granule cell activity at a depth of over 620 μm, through an intact hippocampus during open field behavior. The miniature microscope itself and all supporting equipment are open-source and all files needed for building the scope can be accessed through the Golshani Lab GitHub repository.

Single-photon (1P) epifluorescent miniature microscopes are important tools that have generated numerous neuroscientific advancements for more than a decade[1–8]. These microscopes enable researchers to image neuronal activity from large populations of cells during naturalistic and free behaviors in mice and rats, without the restraint of head fixation imposed by traditional benchtop microscopes. While miniature 1P microscopes are readily disseminated, low cost, and easy to adopt, they have limited optical sectioning capability and increased scattering of visible excitation light by neural tissue, compared to other modalities. In contrast, two-photon (2P) microscopes, focus near infrared (NIR) light to excite fluorescence within a single diffraction-limited volume, which is scanned in time to form an image. This approach provides greater optical sectioning and can resolve fine cellular structure even hundreds of micrometers into

tissue[9–11]. Several groups have designed and developed transformative miniaturized multiphoton microscopes to resolve dynamics from neural populations during free behavior in rodents[12–21]. These microscopes utilize optical fibers to deliver ultrafast laser pulses to a miniature headpiece, and a method of scanning the light across the brain. Scanning is accomplished either via an onboard micro-electrical-mechanical-system (MEMS) scanner mirror, through transmission over coherent optical fiber bundles, or through movement of the distal tip of the excitatory optical fiber. In either case, 2P microscopes need to spatiotemporally focus the excitation light with sufficiently high numerical aperture (NA) optical trains to generate efficient 2P excitation without causing thermal disruption or damage to the tissue being studied. These microscopes relay collected fluorescent signals to detectors positioned either on the headpiece itself or remotely via a

[1]Department of Neurology, UCLA School of Medicine, Los Angeles, CA, USA. [2]Interdepartmental PhD Program for Neuroscience, University of California, Los Angeles, CA, USA. [3]Department of Neurobiology, University of California, Los Angeles, CA, USA. [4]Department of Psychology, The Ohio State University, Columbus, OH, USA. [5]Integrative Center for Learning and Memory, University of California, Los Angeles, CA, USA. [6]Department of Psychiatry and Psychology, University of California, Los Angeles, CA, USA. [7]Department of Neurosciences, University of California, San Diego, CA, USA. [8]Semel Institute for Neuroscience and Human Behavior, University of California, Los Angeles, CA, USA. [9]Intellectual and Developmental Disability Research Center, University of California, Los Angeles, CA, USA. [10]Greater Los Angeles VA Medical Center, Los Angeles, CA, USA. [11]Present address: Picower Institute of Learning and Memory, Massachusetts Institute of Technology, Cambridge, MA, USA. ✉e-mail: pgolshani@mednet.ucla.edu

single, or a bundle of optical fibers. Despite this progress, open-source high performance 2P miniature microscopes that are straightforward to assemble and use at a cost below $10,000 USD do not yet exist.

Here we describe the UCLA 2P Miniscope, an open-source miniature 2-photon microscope capable of recording calcium dynamics from neurons in deep structures and from dendrites over a 445 μm × 380 μm FOV.

## Results

### UCLA 2P miniscope design and performance measures

In this work, we describe the development of the UCLA 2P Miniscope, an open-source miniature 2P microscope that weighs ~4 g and is designed to be easily implemented by neuroscientists. The optical path of the system was simulated using Zemax (Fig. 2), the mechanical housings were built in SolidWorks 2020 (Fig. 1), and the electronic system was developed in KiCAD 7.0 (Fig. 6). We designed the microscope around low-cost, simple-to-fabricate spherical optical components in that are tolerant to misalignment. Almost all optical elements, with the exception of the objective lens and tube lens are available off-the-shelf and shown in Fig. 2. The mechanical components of the system (shown in Fig. 1 were designed to be lightweight, while maintaining a high degree of experimental durability. The housings are also designed with efficient assembly in mind, with highly accurate internal stops, and arrowhead features to allow users to visually confirm correct lens placement and orientation. Additionally, the housing components are all 3D printed, using easily attainable resin-based SLA printers. As such, these components can be made in-house by labs at institutions with maker spaces or inexpensively manufactured by online 3D printing services. Mechanical parts can also be fabricated in large batches, increasing ease of adoption by users. Two onboard silicon detectors enable rapid, high sensitivity fluorescence collection on the head of the animal across two color channels; the high light collection efficiency of these detectors allows the user to acquire high SNR images at large depths through scattering tissue. We also designed interface electronics to allow the user to control the microscope in a straightforward way. All together, these easily assembled

microscopes can record from a 445 μm × 380 μm field of view (FOV) with high resolution (submicron lateral, ~10 μm axial) deep into brain tissue, for low cost compared to both conventional and other miniature 2P microscope systems.

Once the microscope was constructed and assembled, we resolved fluorescent microspheres (Invitrogen, TetraSpeck Fluorescent Microspheres Size Kit, T14792) of various diameters to tune parameters like drive waveform frequency amplitude, bidirectional phase offset, and clock filter frequencies. The FOV was measured by displacing a 4 μm bead (T14792, position #1) from one edge of the image field to the opposite edge using an electronically controlled linear stage with high precision and digital readout. Measurements were taken for each independent lateral axis. This was measured at 445 μm × 380 μm. During free behavior experiments, the FOV was reduced to ~308 μm × 308 μm for reliability. More image parameter details can be found in Supplementary Table 1. We then assessed the axial range of the ETL with a piezoelectric actuator that was calibrated using a commercial z-stage from a benchtop 2P microscope system. Digital commands were sent to the ETL, thereby offsetting the focal plane in depth, and the piezo was adjusted manually to bring the focal plane back to the original view. Required displacements of the piezo to restore focus were recorded for each ETL set point, resulting in a total displacement of 150 μm.

Once the basic imaging parameters were assessed using 4 μm microspheres, the optical resolution was characterized with sub-diffractive 200 nm fluorescent microspheres (T14792, position #4). Full-width-at-half-maximum of the gaussian fits to bead cross-sections were used to describe the resolution of the microscope, which was found to be 980 nm in the lateral dimension, and 10.18 μm axially. These numbers are in close agreement with our optical simulations and suggest an effective excitation NA of ~0.36[22] which matches the predicted NA of 0.36.

Following PSF measurements, ex-vivo tissue slices expressing GCaMP6f were visualized, in order to assess detection sensitivity as a function of excitation laser power. Slides were imaged under ~30 mW of laser power, clear cell bodies and processes were observed,

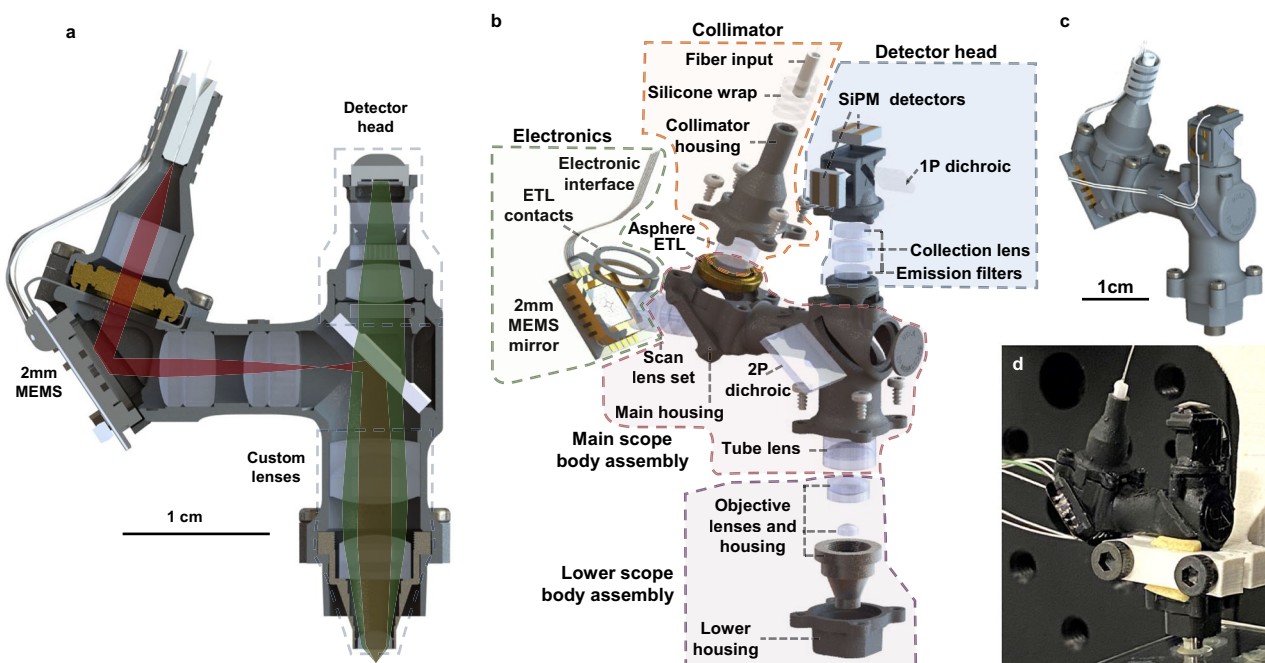

**Fig. 1 | UCLA 2P Miniscope mechanical hardware. a** Optomechanical design of the system, including all housings, lenses, optical filters, and on-board PCBs in cross-section. The red beam denotes the 920 nm excitation, and the green beam path describes the fluorescent collection onto the detector head. **b** Exploded view of the UCLA 2P Miniscope, describing the various sub-assemblies that work together, and the individual parts that comprise them. **c** Mechanical model of the microscope showing external appearance once assembled. **d** Assembled microscope under test at UCLA.

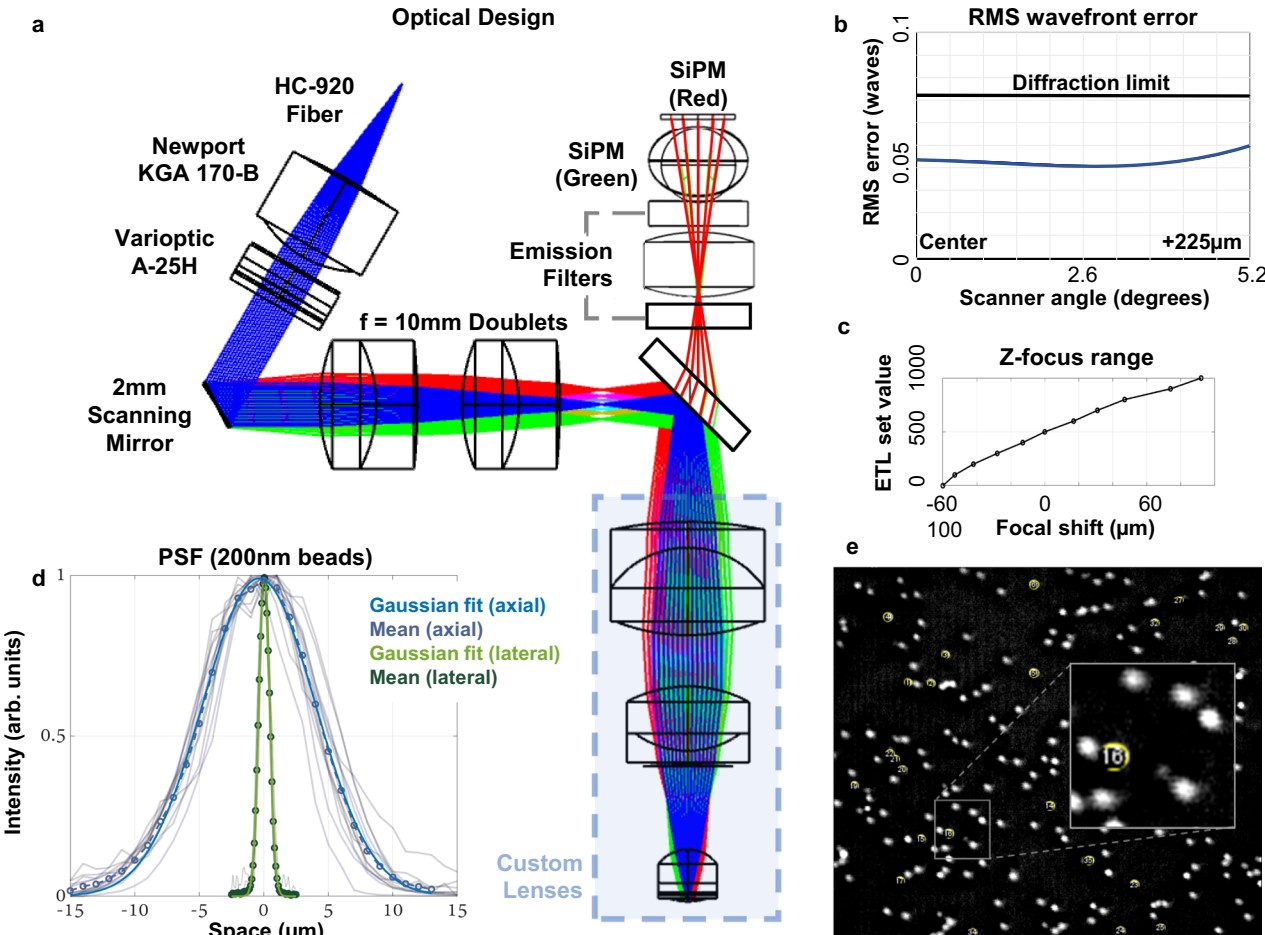

**Fig. 2 | UCLA 2P Miniscope optical system design and performance evaluation.** **a** Full-system optical simulation and optimization in Zemax. All lenses, mirror surfaces and filters were simulated to ensure high optical performance once constructed and assembled. The detector path was simulated separately and superimposed here to demonstrate the full system as a combination of both optical paths. **b** RMS wavefront error over the 2 mm MEMS scanner mechanical range was assessed using a linear Zemax model. System is diffraction limited for an input beam of 1.9 mm over the FOV. **c** Measured axial focus range provided by the electro-tunable lens. **d** Measured lateral and axial point-spread function (PSF). Lateral PSF is 980nm FWHM, axial is 10.18 μm. **e** Example frame when imaging large, 4 μm microspheres for calibration purposes.

confirming that the microscope can generate sufficient signal from physiological concentrations of fluorescent molecules of interest. Following tissue slice recording, we tested the imaging ability of the microscope in-vivo, with head-fixed animals expressing various GCaMP varieties, including 6 f, 7 f, and 8 f. Animals were implanted with a cranial window over cortex or a canular implant above hippocampus (described in detail in the Methods). We then compared image fields resolved with the 2P miniscope to the same regions recorded on a benchtop commercial microscope, described in Supplementary Fig. 5.

**Resolving calcium dynamics in freely behaving animals**
Before resolving images in freely behaving animals, mouse behavior was assessed to ensure that the head mounted microscope did not significantly alter the behavioral state of the animal during 30 min of free foraging. Mice were freely able to navigate an approximately 38 cm × 28 cm chamber for the duration of the recording while sucrose pellets were supplied once every 120 s at a random location throughout the behavioral chamber. The position and locomotion of the mouse were recorded by a behavioral camera positioned roughly 50 cm above the chamber recording at 30 frames per second (FPS). We recorded animal locomotion in both control conditions (without the head mounted microscope) and during active imaging in the dentate gyrus. Positional information of the animal was extracted using

DeepLabCut[23]. Speed and total distance traveled over the session were calculated with a MATLAB script. We found that animals wearing the UCLA 2P Miniscope did not significantly alter the total accumulated distance they run or the speed at which they move over the course of a 30-min recording session compared to those without the microscope (Supplementary Fig. 1).

Following behavioral validation, we resolved place cell activity in dense neuronal populations in hippocampal area CA1. The optical sectioning ability of 2P microscopy allows imaging of activity from dense neuronal populations with high spatial resolution. To understand how the 2P miniature microscope performed at imaging dense populations, we recorded neurons expressing GcaMP7f in the pyramidal layer of CA1. We recorded from populations of neurons expressing Ca²⁺ indicators while the mouse freely navigated the same behavioral chamber used in assessments over periods ranging from 24 to 60 min. Positional information was captured in the same way as the behavioral validations. Image data from the miniature microscope was processed using Suite2P[24] to perform non-rigid motion correction, segment individual neurons, and extract calcium traces. On average during 6 free-behavior recording sessions, we resolved dynamics from 110 ± 8 active neurons in CA1 within the field of view, after processing the movies with Suite2P. The exact parameters for the analysis are included with the rest of the control software, in the form of an.ops file that can be easily referenced or deployed by users. Figure 3

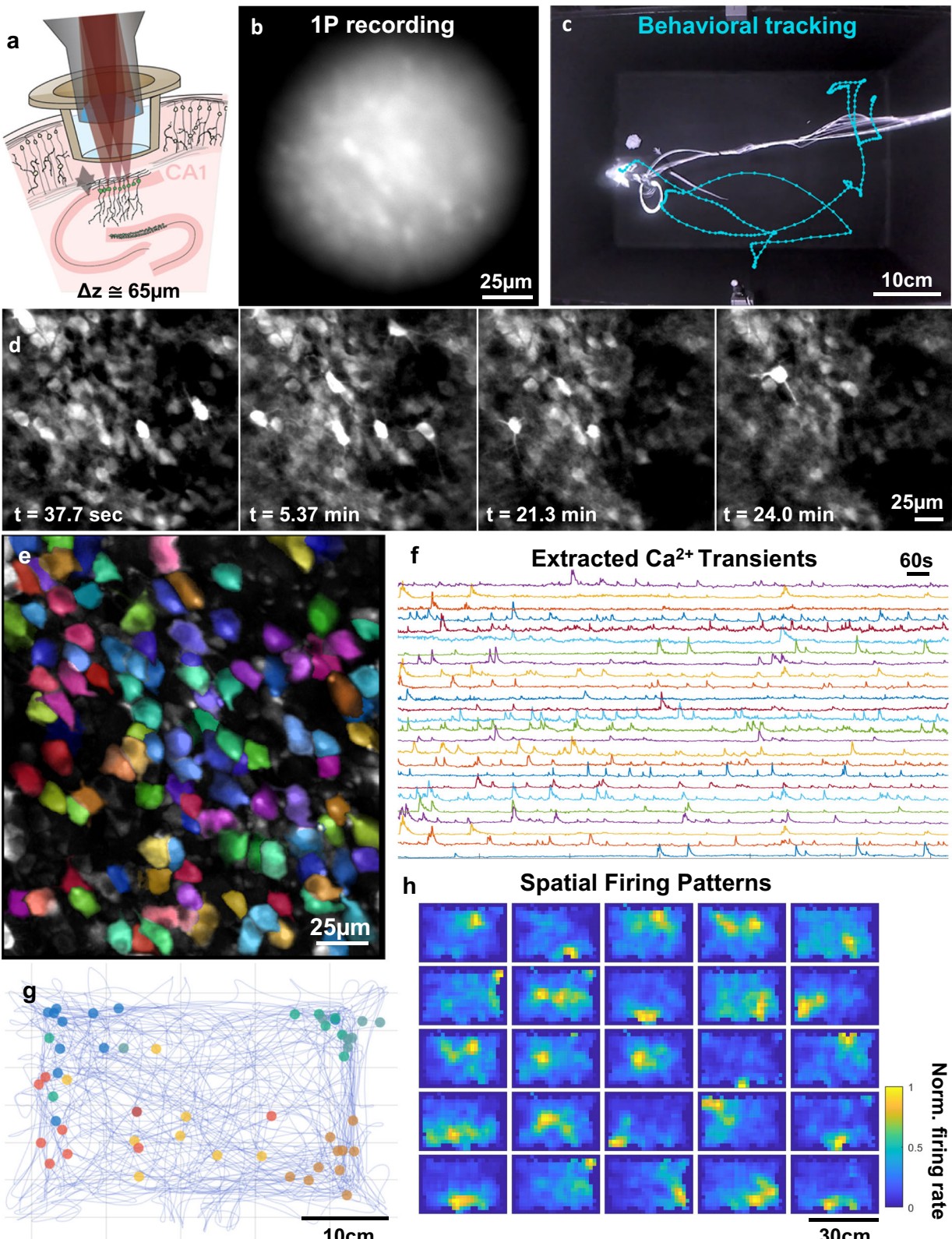

**Fig. 3 | Example CA1 recording session. a** Schematic drawing of the imaging conditions, including the titanium cranial window implant and the objective lens from the UCLA 2P Miniscope. **b** 1P image collected in the same animal, in the same brain region on the same day as all other panels except E. 1P Image was collected with a custom-made benchtop 1P microscope with a 0.5 NA and high resolution scientific Complementary Metal Oxide Semiconductor (sCMOS) image sensor. **c** Experimental mouse in the behavioral chamber, during the >20-min imaging session. Light-blue line shows a subset of the animal trajectory in the chamber over time. **d** Imaging results from the microscope system over the course of the free behavior experiment. **e** Extracted footprints of the active neurons within the FOV (randomly-colored) following motion correction, overlaid on the maximum intensity projection image. **f** Neuropil subtracted activity from the neurons in (**g**, **h**). **g** Firing locations of a subset of neurons from (**f**, **h**) plotted within the behavioral arena. **h** 25 example statistically significant place cells (single sided, multiple comparison corrected, based on Shannon Information content[36]).

demonstrates an example CA1 recording session where an ~20 g female mouse can be seen navigating throughout the chamber while neural activity is recorded. Over the 20-min session, the median speed of the mouse was ~4.1 cm/s which is similar to results from another paper which gauged the impact of microscope weight on median animal speed[17] when considering the system weighs ~4 g. Within one recording session, we observed more than 32 cells with spatial firing preference (highlighted in E, F, G, H of Fig. 3), suggesting that the UCLA 2P Miniscope is able to resolve the activity of place cells in CA1 over behaviorally relevant timeframes. A compiled video from the recording described in Fig. 3 can be viewed in Supplementary Movie 1: CA1 Recording.

Next, we resolved dendritic calcium transients using our 2P miniaturized microscope. 1P miniature microscope systems are largely constrained to studying activity patterns from relatively superficial somas near the implanted window or GRIN lens mainly due to scattering and fluorescent background in neural tissue. Two-photon systems, on the other hand, are able to resolve calcium events in fine cellular structures such as dendrites and axons, hundreds of microns below the surface of the brain. One major goal in the development of our miniature 2P microscope was to build an imaging system with sufficient resolution and sensitivity to study dendritic patterns of activity, since the mechanisms of the integration of neural inputs during free behavior are largely understudied in the field. To assess the system's ability to detect activity from dendrites, we recorded AAV-driven GcaMP6f in layer 2/3 neurons in retrosplenial cortex (RSC) during free behavior on the first color channel, along with cFos-driven mCherry on the second detector channel. Because RSC is such a medial structure, cortical layers are rotationally oriented such that the apical dendrites of pyramidal cells are within the same imaging plane as the cell bodies. We leveraged this anatomical feature to our advantage to study calcium dynamics in somata and apical dendrites simultaneously during free behavior.

To gain optical access to RSC, a 4 mm × 4 mm cortical window was placed over the region (details in Methods). While this window was larger than optically required, it provided experimental flexibility and the ability to locate an FOV where dendrites and somas are both coplanar and clearly visible. Figure 4 shows the results of an RSC experiment, including the image data, mouse position, and activity traces from both cell bodies and dendrites. The high degree of correlation between the activity of the RSC somas and proximal regions of apical trunk dendrites suggests that the large majority of the events we recorded were global calcium events activating both soma and dendrites in the FOV. We also compared the kinetics of somatic and dendritic calcium events during free behavior in Fig. 4d, f. Over 30-min recording sessions, individual dendrites remained stable within the image field (especially in the z dimension) with no need to remove any portions of the recording where dendrites were lost due to motion. The impact of movement is quantified in Supplementary Fig. 6 through monitoring the stability of the mCherry ROI mean intensity over the course of the recording, showing minimal coordinated variations in intensity over time. Optical cross-talk between image channels was mitigated through the use of linear unmixing (coefficient was calculated to be 0.318 between green and red channels). All considered, the UCLA 2P miniscope is well suited for studying dendritic and potentially axonal dynamics during free behavior in mice. For additional information, please view the compiled video from the recording described in Fig. 4 as Supplementary Movie 2: RSC Recording.

Lastly, we resolved calcium dynamics of dentate granule cells, through an intact hippocampus during free behavior. The dentate gyrus (DG) is a critical region within the hippocampal formation and tri-synaptic circuit with extensive and direct connections to other hippocampal subfields and the entorhinal cortex. The principal cells of the DG, dentate granule cells, have been implicated in spatial navigation, episodic memory formation, and discrimination learning[25]. The DG is also a key site for adult neurogenesis in the mammalian brain, including humans[26–28]. Studying dentate granule cell activity in vivo through electrophysiological methods has proven challenging due to their highly sparse but highly salient activity[29], which complicates accurate spike sorting in these neurons. More recently, groups have turned to calcium imaging, which directly visualizes spiking neurons, in order to study the dynamics of granule cells during spatial navigation. However, the current use of calcium imaging in the DG still has significant limitations. In the dorsal hippocampus of the rodent brain, the DG sits deep to CA1 and other subfields within the tri-synaptic circuit. Ideally one would like to maintain the integrity of all the hippocampal subfields and their connections when measuring activity in the DG. The depth of the dentate granule cell layer lies (~500–650 μm) below the surface of the hippocampus (alveus), precluding the use of single photon methods to image the DG within the intact hippocampus. Researchers have relied on the use of GRIN lenses and aspiration and disruption of the overlying CA1 to gain optical access to the DG and resolve dentate granule cells using 1P miniature microscopes[30]. Capitalizing on the ability of two-photon microscopy to image several hundred micrometers into tissue, others have conducted two-photon imaging studies in head-fixed animals using surgical approaches that maintain the structure of the full hippocampus[31]. Such studies rely on virtual or floating environments enriched with tactile, textural, or auditory cues to study spatial navigation[32], but lack key vestibular input that can modulate place cell activity[33]. To address the methodological challenges associated with resolving activity in the DG, we tested the ability of the miniature 2P microscope presented here to resolve granule cell activity in freely behaving mice exploring a novel environment. Sparse dynamics were captured from ~50 granule cells over the approximately 20-minute recording interval. Figure 5 displays the imaging results from an example imaging session, as well as extracted neuronal footprints and activity traces. Additionally, we have recorded the same populations of granule cells over multiple days during free behavior and quantified the spatial tuning of an example neuron in Supplementary Fig. 2. Taken together, these results demonstrate the capacity of the UCLA 2P Miniscope to resolve calcium transients even from deep neuronal populations during free behavior. For additional information, view Supplementary Movie 3: DG Recording, a video from the recording in Fig. 5.

## Discussion

We developed the UCLA 2P Miniscope to be cost effective without significantly compromising performance. Our entire microscope headpiece can be built for less than $7.5k USD currently with low production numbers and likely with scaling can be reduced even further. A full cost breakdown of the head mounted microscope is shown in Table 1. The majority of the cost comes in the form of the custom optical assemblies which are assembled, coated, and tested individually by Optics Technology in New York, USA. With scale, we predict these assemblies to drop in cost and further lower barriers of entry to labs hoping to use miniature 2P microscopes to study neural dynamics in freely behaving animals. To our understanding, this is the first miniature 2P microscope that is open sourced and able to be built for less than $10k USD. All components needed to operate the full system, including the laser, control electronics/computer, and optomechanical parts are listed in Table 2.

We developed the UCLA 2P miniscope as an open-source, easy to assemble miniature multiphoton microscope with high resolution and frame rates optimized for calcium imaging experiments. From the start of the project, lowering cost and maximizing ease of user dissemination has been a critical focus, and we have carried these priorities into each aspect of the microscope design. Custom-made lens assemblies are limited to a modest number of lens elements, for a high degree of mechanical tolerance during assembly and general affordability. The majority of components are off-the-shelf components and easy to

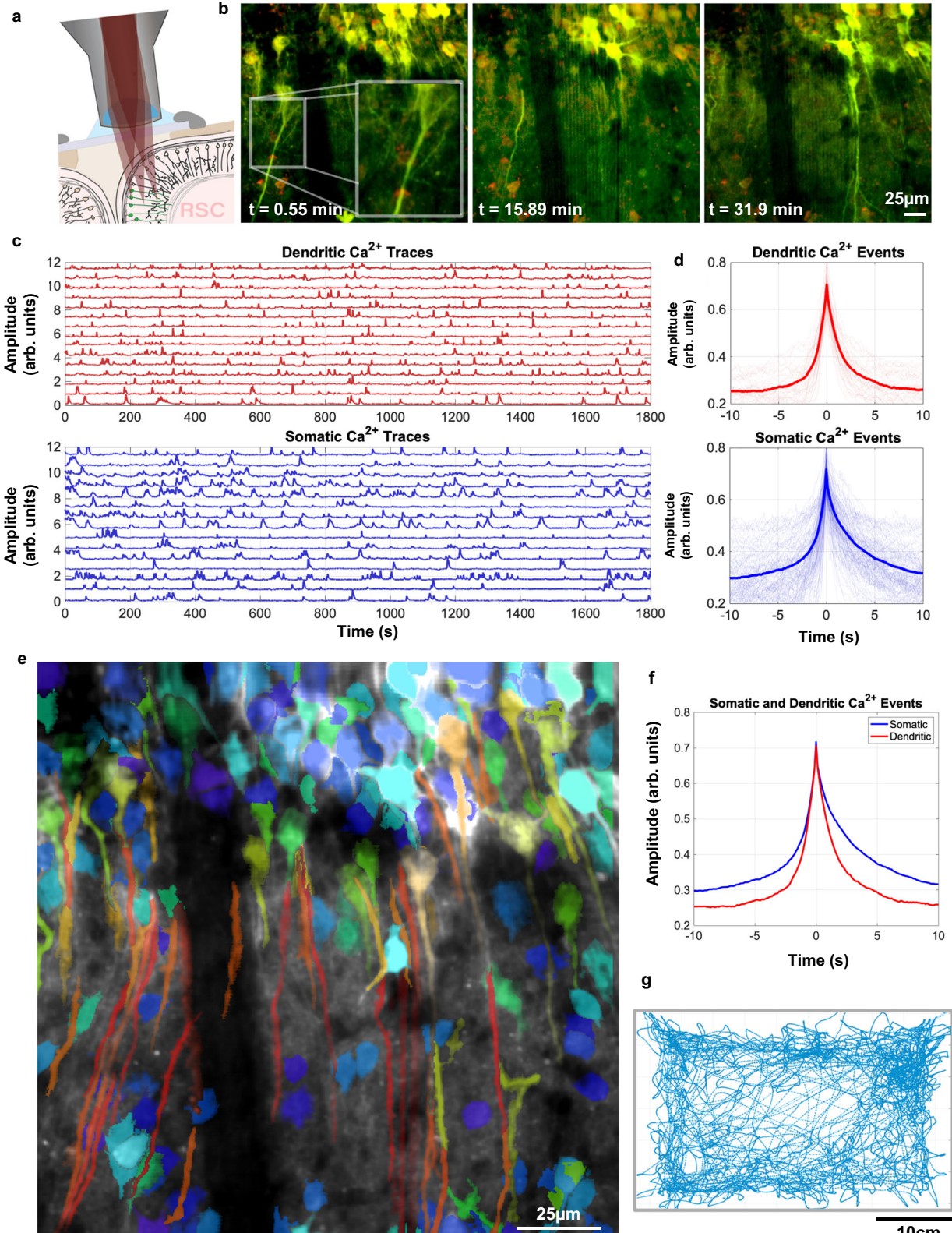

**Fig. 4 | Dendritic imaging in cortex during free behavior. a** Schematic drawing of the experimental preparation used to record from RSC and the curvature of the cortical column that makes resolving long dendrites in the cortex possible. **b** Time-course of still frames over the course of the 30-min recording, highlighting the axial stability of the system and ability to reliably track single projections over substantial timeframes. Axial stability is assessed in more detail within Supplementary Fig. 6. **c** Calcium signals from somatic (blue) and dendritic ROIs (red) plotted separately. **d** Aligned and averaged calcium events from dendritic and somatic ROIs. **e** Pseudo-colored image exported from Suite2P which colors ROIs by aspect ratio. This is superimposed on the mean of the entire recording. **f** Overlay of mean somatic and dendritic calcium dynamics from a single calcium event. **g** Animal trajectory over the 30-min experiment.

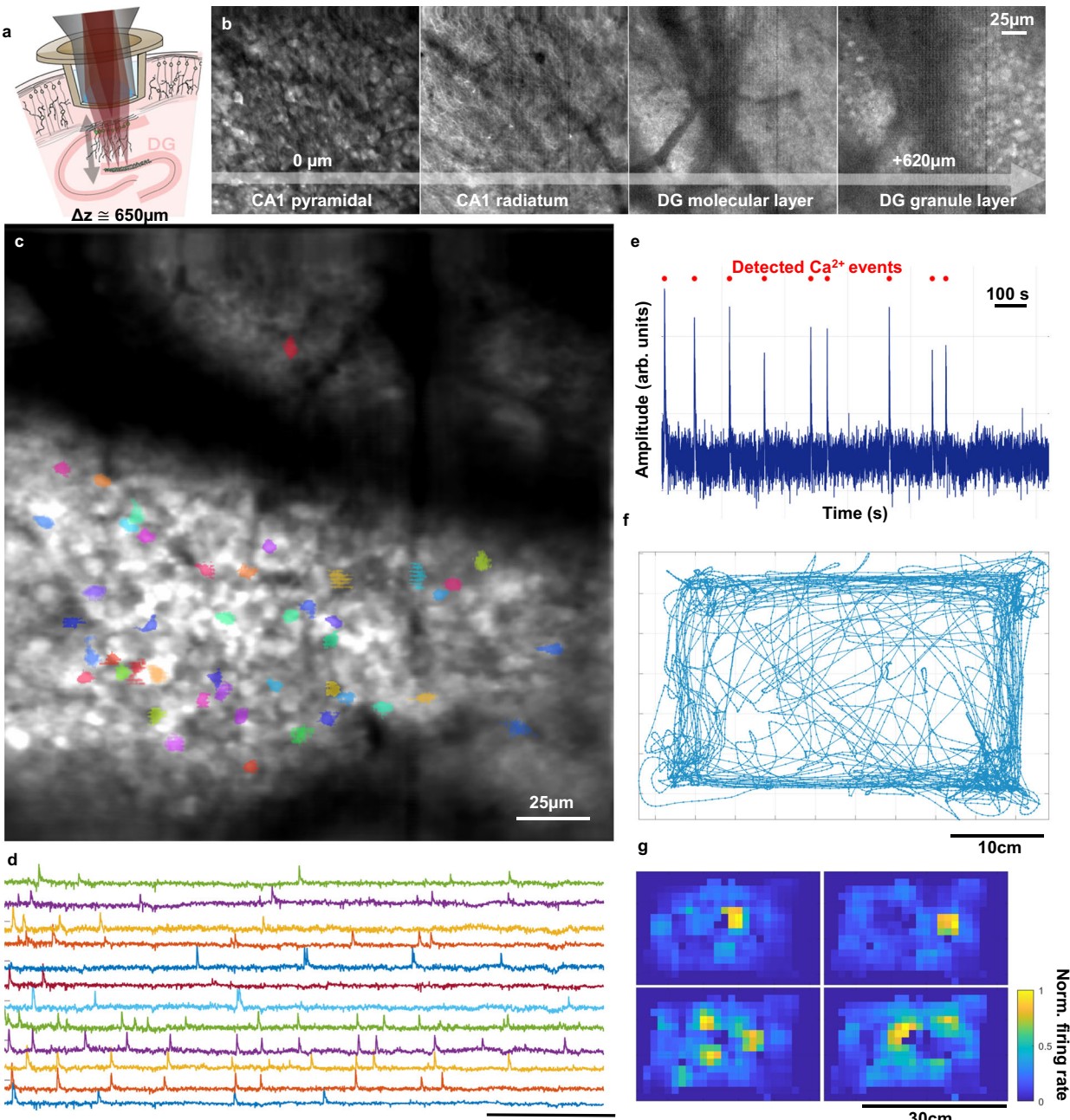

**Fig. 5 | Recording calcium dynamics in deep structures with the UCLA 2P Miniscope. a** Schematic drawing of the titanium implant relative to the image field in DG. Cells in CA1 were also expressing GCaMP8f. **b** Montage of frames from a continuous z-stack in a head-fixed mouse expressing GCaMP8f. The microscope was able to continuously image from CA1 to the granule cell layer of DG. **c** FOV from DG, through an intact hippocampus during a 20-min free behavior experiment. Colored cells were identified as active using Suite2P. **d** Activity traces from a subset of the identified active neurons shown in (**c**). **e** A single example neuron activity trace. Calcium peaks identified from the deconvolved signal plotted as red dots above the neuropil-subtracted activity. **f** Animal trajectory over the 20-min experiment. **g** Statistically significant (single sided, multiple comparison corrected, based on Shannon Information content[36]) spatial activity plots for a subset of the cells displayed in (**c**).

procure, in order to lower barriers of entry to labs interested in using this technology. The housings are made using low-cost resin-based 3D printers (all files are uploaded to the Git repository, including.form files) and can be made easily in large batches. We integrated silicon-based detectors that have been shown to be effective in large microscope 2P systems[34] onto the collection head of the microscope. A 3-photon miniature microscope system in the literature for and mice[21] has also demonstrated high performance with these detectors, and removes cumbersome and photonically-lossy fiber bundles from the assembly (specifics of fiber bundle losses in Collection Efficiency). In addition, we designed electronic interfaces to be plug-and-play and easy to interface with. Initial, fully functional versions of these electronics are uploaded to an open GitHub repository. The result is a ~ 4 g miniature 2P microscope system with sub-micron lateral resolution over large image fields, useful for studying neural dynamics of deep structures or fine features in freely behaving animals.

Our goal is to make miniature 2P microscopy as accessible to as many users as possible, while pushing the technical limits of what such systems are capable of achieving. Intentional design tradeoffs were made throughout the engineering process which ultimately prioritized

**Table 1 | Cost breakdown for the UCLA 2P Miniscope microscope headpiece**

| Component | Manufacturer, Part Number | Quantity | Cost | Subtotal |
|---|---|---|---|---|
| Objective Lens | OT, Custom – 1 | 1 | $4000.00 | $4000.00 |
| Tube Lens | OT, Custom – 2 | 1 | $1100.00 | $1100.00 |
| Scan Lens | ThorLabs, AC050-010-B | 2 | $51.38 | $102.76 |
| Aspheric Lens | Newport, KGA170-B | 1 | $104.00 | $104.00 |
| Collection Lens | Edmund, 47-895 | 1 | $62.50 | $62.50 |
| 2P Dichroic | Chroma, ZT775sp-2p | 1 | $540.00 | $540.00 |
| 1P Dichroic | Chroma, T550lpxr | 1 | $112.50 | $112.50 |
| 2P Emission Filter | Chroma, ET750sp | 2 | $150.00 | $300.00 |
| MEMS Scanner | Mirrorcle, A7M20.2-2000AL | 1 | $591.00 | $591.00 |
| Electrotunable Lens | Varioptic, A-25H1 | 1 | $135.00 | $135.00 |
| Mechanical Housings | Custom Made | 4 | $10.00 | $40.00 |
| PCBs | PCBWay, Custom Made | 1 | $50.00 | $50.00 |
| SiPM Detectors | Hamamatsu, S13360-3075PE | 2 | $52.88 | $105.76 |

These costs are as of manuscript submission when components were purchased at small scale, total of all components together is $7245.52 USD. Please note that these expenses do not include key additional hardware, such as laser and traditional 2P microscope acquisition electronics.

**Table 2 | Complete cost breakdown for the UCLA 2P Miniscope system and all associated components**

| Sub-Assembly | Manufacturer | Quantity | Cost | Subtotal |
|---|---|---|---|---|
| Microscope Headpiece | UCLA 2P Miniscope, Custom | 1 | $7244 | $7244 |
| Multi-photon Laser | Coherent, Axon 920 1W, TPC | 1 | $63,500 | $63,500 |
| DAQ Electronics and Computer | MBF Bioscience | 1 | $31,500 | $31,500 |
| Bench-Top Laser Optics | ThorLabs | 1 | $3000 | $3000 |
| Interface Electronics | Digilent, Hamamatsu, Custom | 1 | $2000 | $2000 |
| HC Optical Fiber | NKT Photonics | 2 | $1200 | $2400 |
| Control Software | MBF Bioscience, ScanImage (Free) | 1 | $0 | $0 |

These costs reflect purchase prices in the past and may be slightly different currently. Total of all components together is in $109,644 USD and represents the approximate cost of setting up the miniature 2P microscope system with no existing equipment besides an optical table.

user adoption over absolute performance. One example of this is the tunable lens. Other microscope systems use stacks of piezoelectric tunable interfaces to generate sufficient optical power to translate the focal plane by respectable amounts[17]. While these elements are lightweight (0.06 g) and seem to perform very well, they are difficult to align together into a colinear, stacked assembly and install into the microscope body, resulting in a barrier to entry for users. In contrast, we use off-the-shelf Varioptic lenses which are substantially heavier (0.36 g) yet are easy to source and install. This intentional decision was made not due to performance, but to achieve increased user friendliness and adoption capability. We believe that these benefits outweigh the cost associated with the modest increase in weight of the UCLA 2P Miniscope compared to other similar systems. In the same spirit, we provide assembly guides on the Git repository to describe the construction of these microscopes in great detail and describe the full process of taking parts from the 3D printer, processing them minimally, and building them into a functional UCLA 2P Miniscope.

The UCLA 2P miniscope also uses on-board Si-based detectors, similar to those used in 3 P applications[21]. Such a detection scheme substantially improves collection efficiency of the microscope and thus makes resolving dynamics from deep structures like DG possible, at the expense of added weight. The microscope in its current form uses two detectors, to record from red and green channels, as well as collection optics and spectral filters to isolate colors onto discrete detectors. Because of this, the microscope incurs some additional weight, at the benefit of being able to record deep and challenging brain areas in two colors. To minimize weight, the detector assembly does not include discrete emission filters, instead we use linear

unmixing (coefficient = 0.318) after image collection to achieve multicolor imaging while mitigating cross-talk.

Initial designs of the UCLA 2P Miniscope used a flexible bundle of optical fibers, such as those demonstrated in the literature[15–17,19] to relay fluorescence to a set of GaAsP PMTs on the optical bench. Such prototypes used a highly flexible optical fiber bundle from SCHOTT (AO-ERP Part No. IB1651350) with a 1.65 mm core (1.45 mm quality area) and 1.35 m length. The transmission efficiency of this bundle was measured as 30.96% (±0.36%) over 6 measurements using a 532 nm laser diode passing through two adjustable apertures to ensure underfilling of the quality area. The power transmission through the remaining collection optics for the prototype setup was also measured, including propagation through a NIR laser blocking filter (Chroma ET750-SP, same as the UCLA 2P Miniscope), a 1P dichroic mirror to split green and red channels (Chroma T550LPXR, same as the UCLA 2P Miniscope) and an emission filter (Chroma, ET510/80 M, UCLA 2P Miniscope does not use an emission filter due to size, thickness and weight reasons). Collected fluorescence also transmits through two 16 mm focal length, high NA condenser lenses (ThorLabs, ACL25416U) on the prototype benchtop detector. When measuring power at the location where the GaAsP PMT was typically located. These measurements show that ~50% of the collected signal is lost through the benchtop collection optics, putting the efficiency of the complete setup at 15.77% ± 0.28% over 6 measurements.

In order to compare results with the SiPM based detector scheme, we repeated these measurements but placed the detector housing described in Methods immediately before the position of the fiber bundle. Optical power of the 532 nm laser following the two adjustable

apertures was measured again, immediately before entering the detector housing and at the position of the SiPM following all filters and dichroic beam splitters, resulting in a transmission efficiency of 71.39% ± 0.55% over 5 measurements. Based on these measurements we conclude that having detectors located on the head of the animal result in substantial gains in fluorescence collection efficiency, resulting in the capacity to measure dim signals deep in the brain.

Lastly, the optical design is rather simple, and the mechanical housings are intentionally robust. We know from experience that imaging devices used in freely behaving animal experiments are subject to great abuse, during which housings can break or optical misalignments may occur. By simplifying the number of components and interfaces, we not only made the optical tolerances forgiving, but we also made the device reliable. Despite this simplicity, the microscope is able to resolve sub-micron features in neural tissue. In key portions of the mechanical housings, the wall thickness is deliberately thick, resulting in components resilient to knocks, bumps, and drops. This increases weight but ultimately improves useability by researchers and ensures the device works in realistic environmental conditions, which is of paramount importance. Altogether, the UCLA 2P miniscope is a high-performance microscope intended to be set up and used by other labs. Decisions were made throughout the design process which included known sub-optimizations, which in turn impose limitations on performance. In each circumstance, these tradeoffs were made to the benefit of user adoption and reducing the barriers to entry as much as possible.

The development of miniaturized multiphoton microscopy methods for recording neural circuit activity in freely behaving animals is a rapidly advancing subfield at the intersection of neuroscience and optics. Over the past several years, many transformative systems have been developed and reported in the literature, each with a host of advantages and disadvantages. To compare the system designed in this manuscript with others in the literature, we have included Supplementary Table 2 (modified with permission from ref. 35) to compare and contrast the performance of this microscope with others in the field.

The majority of the effort in this project went to the design and development of the head-mounted microscope hardware. While this is an critical aspect to the overall system, there are other key components which are critical for function and user adoption. Firstly, freespace propagating 2P laser sources are expensive and present a financial as well as technical barrier to many labs. Launching these lasers with multi-axis fiber launches can be complex for users without optical alignment experience. The advent of lower-cost, fixed frequency fiber lasers with direct fiber coupling, and large amounts of GVD compensation has helped bring the challenges down to user adoption. In the future, we hope to work with laser manufacturers to offer solutions which are easily implemented at an affordable price. Secondly, we are currently working on building standalone low-cost electronic drivers to control microscope hardware via an intuitive and open-source python-based GUI. We believe that users should just be able to plug in the microscope over a thin, flexible bundle, start a simple program alongside a control suite like ScanImage, and be able to record quickly and easily in freely behaving animals. The next generation PCBs and control software will be made available as soon as it is completed. Additionally, we aim to make these devices at scale, such that unit costs per microscope can be reduced further. Our aim is to produce microscopes well within the reach of users to ensure that the miniature microscopes can be used by as many labs as possible. Lastly, we would like to expand the technical capabilities of the microscope system in the future, through the integration of new components and mechanical designs. The microscope is currently limited in frame rate due to the resonant frequency-mirror inertia trade-off for electrostatic MEMS scanners, which also limits the excitation NA achievable given the size of the FOV. We believe that

implementation of piezoelectric integrated scanner technologies to further expand the field of view, frame rate, and NA of the optical system would push miniature multiphoton microscopes substantially forward in the future. We also believe that integration with unique lens assemblies, such as axicons[35], diffractive optical elements, or freeform lenses would pose distinct technical advantages that could greatly expand the capabilities of these optical assemblies to sample in even more demanding experimental conditions.

## Methods

All experiments were conducted per National Institute of Health (NIH) guidelines and with the approval of the Chancellor's Animal Research Committee of the University of California, Los Angeles.

### UCLA 2P miniscope system

**Optical Hardware.** We designed the UCLA 2P miniscope to be lightweight, with minimal tethering complexity, and to enable sustained imaging of dynamics from deep structures during free behavior. The microscope's full optical simulation can be seen in Fig. 2a and optomechanical design and beam-path are shown in Fig. 1a. Ultrafast laser pulses from a fixed frequency fiber laser (Coherent, Axon 920-1TPC) are launched to a commercially-available hollow-core photonic bandgap fiber (NKT, HC-920) with a 3-axis fiber launch and a coupling aspheric lens (ThorLabs, C230TMD-B). The light is delivered to the microscope headpiece via the thin flexible fiber featuring a mode field diameter of 6 μm, NA of -0.1 (measured), group velocity dispersion of 90 ps/nm/km, and transmissive loss less than 150 dB/km. Once making it to the microscope's mechanical assembly, ultra-fast laser pulses are collimated with a molded aspheric lens (Newport Photonics, KGA170-B). An electro-tunable lens (ETL) (Varioptic A-25H1) converges or diverges to the collimated beam according to a supplied electrical signal, remotely controlling the depth of the focal plane (nominal WD = 720 μm) of the imaging system by ±75 μm. Following the tunable lens, the light is scanned on an integrated 2 mm diameter 2-D MEMS scanner (Mirrorcle Tech, A7M20.2-2000AL). A set of two 10 mm focal length doublets are positioned near one another to form a scan lens set (ThorLabs, AC050-010-B), with their exact positions defined by hard stops located within the microscope housing. A dichroic beam-splitter (Chroma, ZT775sp-2p-UF1) directs the laser beam, which underfills a custom fabricated 6 mm diameter doublet and 3-element objective assembly (Optics Technology, USA) thereby focusing the light into the brain. The excitation path is designed to be diffraction-limited at the numerical aperture of approximately 0.36 over the 445 μm × 380 μm image field of view. Once 2P excitation is generated, fluorescent photons are collected through the objective utilizing its full NA (-0.6, based on simulation results over FOV), and relayed through the tube lens and dichroic mirror, to a collection head fitted with on-board silicon detectors. The collected light is spectrally filtered to reject laser components using two emission filters (Chroma, ET750sp) and directed with a 4.5 mm focal length plano bi-convex collection lens (Edmund, 47-895). Green and red components are split with another 1P dichroic mirror (Chroma, T550lpxr) and corresponding signal is collected on fast, sensitive, and low-cost Hamamatsu MPPC detectors (S13360-3075PE) located on the collection head.

**Mechanical system.** To support and appropriately orient all of the optical components simulated in Fig. 2, a custom-made mechanical housing was designed in SolidWorks 2021 and fabricated using SLA-based resin 3D printers. The microscope's optomechanical design can be seen in Fig. 1a–c and is composed of four sub-systems. First, a collimator assembly, which supports the optical fiber ferrule and a collimation aspheric lens. The electrotunable lens is positioned between the collimator and the main body of the microscope, secured in place via four thread-forming torx screws. The screws also provide a

clamping force between the electrodes of the UCLA 2P Miniscope flex-PCB and the electrotunable lens, making for a reliable and secure electrical connection. The electronics subsystem, described in detail below, is fixed to the main housing using UV adhesive (Norland 68). The main scope body contains the scan lens set, the custom-made tube lens, and 2P dichroic mirror. The lower scope body assembly secures the custom objective lens to the microscope, and interfaces with the baseplate on the head of the animal. Lastly, the detector head press-fits into the main scope housing and can be secured with a small amount of UV adhesive. After printing, component bores are reamed to high dimensional accuracy using precision tools in a minimal process that can be done quickly and easily, as documented on the assembly guide uploaded to GitHub.

**Electronic system.** Control of all microscope functionality is accomplished with several custom PCBs which relay control signals to one another. Off the animal, there is a signal interface PCB, seen in Fig. 6a, which combines inputs from an I²C controller (NI, USB 8451 OEM), Hamamatsu SiPM drivers, an analog MEMS amplifier (Mirrorcle BDQ PicoAmp, Analog), and other equipment into a single connector that can be easily accessed for experimentation. The miniature microscope

PCB itself, shown in Fig. 6b, connects to the signal interface PCB over a thin and flexible set of coaxial cables. This allows for control of the scanner position, as well as on-board driving of the electrotuneable lens using the I²C data protocol. The circular, flexible electrodes on the PCB wrap around the electrotuneable lens and deliver the appropriate signals from the ETL driver to the lens body. Altogether these PCBs work alongside traditional electronics for controlling benchtop 2P microscopes and enable straightforward and reliable driving of the miniature 2P system.

**Custom control software.** The miniature microscope hardware is mainly controlled by ScanImage (MBF Bioscience, free version) a commonly used software and user interface in the field of multiphoton microscopy. The microscope is driven typically with a raster scanning waveform in bidirectional mode, with a resonant frequency of 1.65 kHz. Imaging frame-rates for the datasets shown varied between 7.92 and 8.62 Hz, with specific parameters detailed in Supplementary Table 1. All machine configuration and user configuration files are available to users for direct implementation and ease of use. Running alongside ScanImage, a custom MATLAB script is used to control ETL position using the NI I²C controller (USB 8451 OEM) fitted on the signal

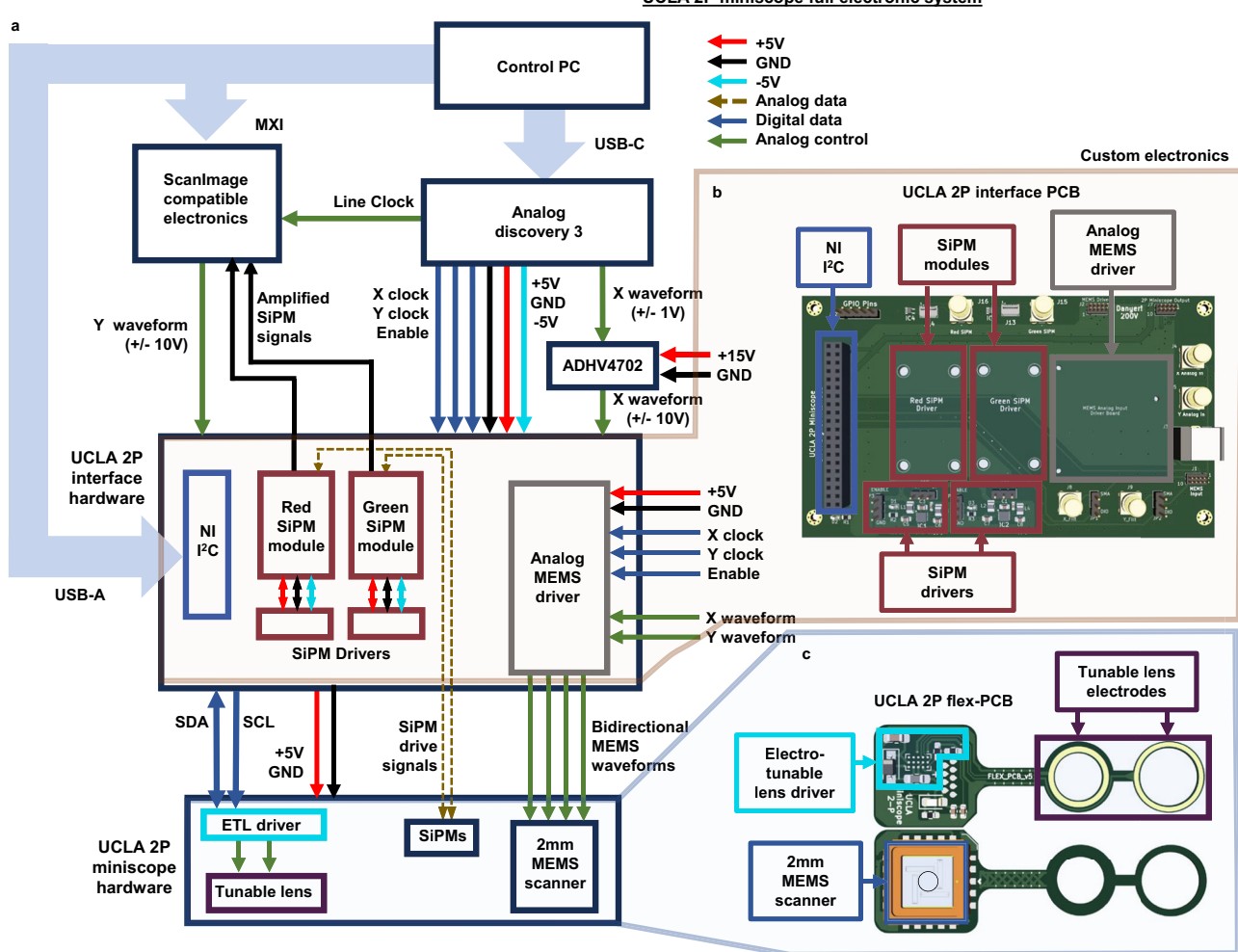

**Fig. 6 | UCLA 2P miniscope electronic circuitry. a** Simplified block diagram of the custom and off-the-shelf electronic components. The custom PCBs communicate with one another via thin coaxial cables and the specifics of the connections can be seen. **b** UCLA 2P Miniscope interface PCB. This single low-cost 2 layer PCB integrates electronic signals from various sources and packages it into a single connector; the microscope plugs into this connector. This interface PCB receives four

daughter boards: two SiPM drive modules (Hamamatsu), an I²C controller (NI), and the MEMS amplifier (Mirrorcle). The interface PCB receives inputs from ScanImage-compatible electronics and additional hardware. **c** UCLA 2P Miniscope flex PCB. These electronics are a fundamental part of the headpiece and are used to control the MEMS scanning mirror and electrotunable lens.

interface PCB, and a free MATLAB hardware support package (National Instruments NI-845x I²C/SPI Interface).

**Resolution measurements.** In order to assess the resolution of the microscope system sub-diffraction diameter beads were resolved. First, the drive waveforms were reduced in amplitude to effectively zoom the FOV to 50 μm in each direction, as verified with the high-resolution translation stage. Then, individual 200 nm microspheres were brought into view with the ETL set to the digital value 500 (no appreciable focal power) and adjusted axially maximize bead diameter. 50-frame image sequences of 512 × 512 resolution were acquired, making the per-pixel sampling rate 97.5 nm in the lateral dimension. Frames were averaged together to reduce noise prior to taking line-profile measurements through the centroid of the bead. For axial measurements, we used a calibrated, high-resolution piezo-electric actuator to displace the imaging slide by ±20 μm about the focal plane over approximately 40 s. The displacement of the slide and the onset of the microscope recording were time-synchronized using external electronics (Digilent, Analog Discovery 3). In the axial direction, 400 images were collected per stack, grouped mean projections of 10 images were calculated, resulting in the final axial measurement stack of 40 frames with an axial displacement of 1 μm between frames. A circular ROI is drawn which encircles a single bead and the mean value of the pixels within the ROI are computed for each image in the stack. The Z-axis profile was used to assess the changes in mean value across the 40 μm displacement, and thus, axial resolution. The final measurement included >10 beads for each direction (lateral and axial) which were then shifted relative to each other as to align their maximum values, accounting for any misalignments in experimental setup. These measurements were then averaged over all beads, and a gaussian function was fit for each direction (Fig. 2d). A subset of measured beads and their lateral and axial profiles can be seen in Supplementary Fig. 7. Through our measurements (and the data presented) we noticed a reproducible low-amplitude striping artifact that we believe to be due to capacitive coupling between the high voltage driving waveform (on the x-axis of the MEMS scanner specifically) and the small-amplitude SiPM signal currents that pass alongside the high voltage drive signal prior to transimpedance amplification. We have included a custom FFT signal filter and built a Fiji macro around it, to selectively remove this high frequency artifact from images. These resources have been uploaded to the GitHub directory as FFT_Filter.tif, and Preprocess_Filter.ijm respectively. Of note, recent design revisions of the microscope feature increased coaxial shielding and thus does not exhibit this artifact.

**Surgical preparation for imaging.** Imaging experiments used to validate the performance of the UCLA 2P Miniscope were conducted on both male and female adult (>P60) mice. Animals were anesthetized with isoflurane during the entire surgical procedure. Exposed fur on the skull was trimmed away in a sterile manner, and animals were thermally maintained using a homoeothermic temperature control blanket and controller from Harvard Apparatus. The animal was secured in a stereotaxic frame and subcutaneously administered local anastatic (lidocaine at 1.5 mg/kg and carprofen at 5.5 mg/kg) 30 min before the scalp was removed using a scalpel. The skull was scraped to facilitate effective bonding between the skull and the implanted optical window. For CA1 and DG recordings, a 3.6 mm circular craniotomy was made in the skull just above PPC using a precision dental drill, taking care as to not damage the underlying dura. Cortical experiments used a 4 mm × 4 mm square craniotomy. Following the removal of the section of skull, the site was flushed with cortex buffer (NaCl = 7.88 g/L, KCl = 0.372 g/L, HEPES = 1.192 g/L, CaCl2 = 0.264 g/L, MgCl2 = 0.204 g/L, at a pH of 7.4) until all bleeding subsided. A viral injection of AAVs to express GCaMP6f, GCaMP7f or GCAMP8f was

completed at a rate of 1nL/s into the targeted structure using a Nanoject injector (Drummond Scientific). For hippocampal CA1 recordings, a unilateral 1000 nL injection of *pGP-AAV-syn-jGCaMP7f-WPRE* (Addgene: 104488) was performed. Dentate gyrus recordings used 500nL of *pGP-AAV-syn-jGCaMP8f-WPRE* (Addgene: 162376) delivered unilaterally. Animals used for RSC imaging received a mixture of *AAV1.Syn.GCaMP6f.WPRE.SV40* virus (Addgene: 100837), *AAV1.cFos-tTA* and *AAV1.TRE.mCherry* at 20 to 120 nL/min into dorsal cortex using stereotactic coordinates −1.7 and −2.3 mm posterior to bregma, 0.5 mm lateral to midline, and −0.8 mm ventral to the skull surface. For cortical experiments, a thin #0 optical cover-glass (4 mm × 4 mm) was placed onto the surface of the brain and secured to the skull using cyanoacrylate and dental cement. This results in an exposed area of approximately 3.5 mm × 3.5 mm spanning the midline. When imaging sub-cortical structures like hippocampus or dentate gyrus, overlying cortex was slowly aspirated, taking care to not damage the alveus. A custom-made titanium cannular plug (implanted portion: 3.5 mm diameter × 1.315 mm depth; flange above skull: 5.7 mm diameter × 0.285 mm thick) was fitted with a sterile 3 mm #0 glass coverslip using optical adhesive (Norland 68) and secured using a UV-light gun. This plug assembly is lowered into the craniotomy and secured to the skull with a thin layer of cyanoacrylate glue and dental cement. Any residual tangential space between the surface of the skull and the flat base of the cannular plug was filled using dental cement and allowed to harden completely. In all cases, the animals also have a stainless-steel head-bar fixed to the skull that enables researchers to lower the microscope easily and find an optimal FOV for recording. Following surgery, animals were given carprofen subcutaneously at 5.5 mg/kg every 12–24 h to minimize pain and inflammation over the first 48 h post-op. Animals were also provided amoxicillin-treated water at 0.5 mg/mL concentration over 7-days and were allowed to rest for an additional 7-days to allow for full recovery and to ensure the cover-glass had cleared. After 14 days, expression levels were assessed using a benchtop 2P microscope (Scientifica VivoScope) fitted with a Nikon 16x/0.8NA water immersion objective. Once expression levels and overall tissue health was confirmed, animals received a baseplate. First, the animal was head-fixed and ~1 mL of distilled water is added to the surface of the cranial window. Then, the UCLA 2P Miniscope was lowered over the cover glass, with the baseplate already fitted to the lower microscope housing. The position of the miniature microscope was controlled using a manual 3-axis stage and adjusted while imaging, until an optimal FOV was found. Once the image field was identified, the baseplate was fixed using dental cement and allowed to fully cure. The miniature microscope was then removed, and a small 3D printed window cover, the exact shape of the lower microscope housing, is placed into the baseplate itself, to help mitigate the ingress of dust and debris onto the recording area. This cover is very light (~0.335 g) and was fixed in place with two small screws in the same way as the microscope itself. The animal was then removed from head fixation and returned to its home cage.

**Object exploration during free behavior.** The microscope was attached to the baseplate and animals were allowed to explore a 28 cm × 38 cm chamber containing a food pellet covered with peanut butter which they could consume. Behavior was recorded from a high-resolution webcam (Logitech B910), and calcium signals were collected by the miniature multiphoton microscope over time spans ranging from 10 to 60 min. Behavioral movies and calcium imaging movies were synchronized by an in-frame LED which turns off on the precise onset of the microscope recording. Behavioral frames are aligned to the first miniscope frame.

**Free foraging during free behavior.** The same experimental chamber described above was used in a free foraging behavior assessment over

30 min. Sucrose pellets were supplied once every 120 s at a random location throughout the behavioral chamber and animals were able to freely forage and consume the sucrose pellets. The position and locomotion of the mouse were quantified in the same way as above, both in control conditions and during active imaging of granule cells in DG.

## Analysis methods

**Measuring place firing preference in CA1 and DG.** Microscope images were processed using Suite2P[24] version 9.2 to correct for motion artifacts, perform segmentation, and calculate activity traces over time. Suite2P deconvolved the slower calcium dynamics according to the specific GCaMP being used in a particular experiment, generating a spks matrix used for downstream analysis. Animal locations were calculated from a time-synchronized behavioral video using a trained neural network via DeepLabCut[23]. To evaluate the spatial firing properties of neurons, a MATLAB script was written which calculates the the Shannon Information content for each neuron in the FOV[7]. To do this, position information was down sampled to the same number of time points as there are behavioral frames throughout the experiment. Space within the frames was discretized into 2.08 cm × 2.08 cm spatial bins for CA1 and DG recordings. A speed threshold of 2 cm/s was applied to restrict consideration of data while the animal was immobile. An occupancy matrix was constructed by summing the number of samples spent in each spatial bin, and a spatial neural activity matrix was calculated as the sum of all deconvolved activity for each spatial bin. Each of these matrices were smoothed using a 2D gaussian kernel with a sigma of 6 cm. After filtering, the smoothed spatial neural activity matrix was divided by the smoothed occupancy matrix to arrive at a spatial neural activity rate for each neuron. Shannon Information, ($I$), was calculated for each neuron using the Kullback-Leibler Divergence formula, as implemented in ref. 36.

$$I = \sum_i \left( \frac{\lambda_i}{\lambda} \log_2 \left( \frac{\lambda_i}{\lambda} \right) p_x \right) \tag{1}$$

Such that $\lambda_i$ is equal to the rate of neuronal activity at bin ($i$) and $\lambda$ is the mean neural activity rate[7]. Statistical significance was assessed using circular shifting. Neural activity traces were offset by a random number of samples in time, ranging from one sample to the total number of samples within the recording such that the end of the neural activity vector is circularly wrapped to the beginning. This effectively shuffles the relationship between the animal's position and the extracted neural activity. Information content was calculated for all 500 shuffles, across each identified ROI. Only the ROIs whose information content was significantly greater than chance ($p > 0.95$) were considered place cells and plotted in Figs. 3 and 5.

## Statistics and reproducibility

This work did not use statistical methods to determine sample sizes. Data were collected as a series of preliminary example use cases to demonstrate the capabilities of the imaging system described. As such, experiments were not randomized and were not blinded.

## Reporting summary

Further information on research design is available in the Nature Portfolio Reporting Summary linked to this article.

## Data availability

All of the design files related to the construction and implementation of the UCLA 2P Miniscope are available publically through the Golshani Lab GitHub[37] (https://github.com/golshanilab/UCLA_2P_Miniscope). This also provides links to demonstration data, including behavioral videos and calcium imaging data.

## Code availability

Custom scripts to control the microscope are located in the Control Software portion of the GitHub repository[37] (https://github.com/golshanilab/UCLA_2P_Miniscope) along with a demonstration that includes data processing code to calculate place fields and expected outputs from the scripts.

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

## Acknowledgements

First, we acknowledge and thank Professor Julie Bentley at the University of Rochester Institute of Optics, for her instrumental work designing the custom tube lens and objective assembly. We acknowledge members of the Golshani Lab: Zoë Day: For constant support with orders, animal surgeries, and experiments. Jiannis Taxidis: For early help with free behavior experiments. Samara Miller: For early help with projection imaging. We acknowledge members of the Aharoni Lab: Federico Sangiuliano Jimka: For endless help with electronics and troubleshooting. Takuya Sasatani: For electronic design help and discussions regarding SiPM drivers. We acknowledge Collaborators: Alipasha Vaziri: For direction and support throughout the project. Tobias Nöbauer: For many beneficial optical discussions and brainstorming sessions. João Couto: For guidance, inspiration and help through many steps of the process. Bruno Pichler: For discussions around control electronics and thinking to use Digilent devices. Funding Sources: NIH: T32MH073526 to C.D., F31MH123111 to B.M., U01NS128664 to P.G., D.A, A.S.J., and M. Shtrahman.

## Author contributions

B.M. M.Shtrahman. A.J.S., D.A. and P.G. conceived the imaging system. B.M. designed, fabricated, and built the system with input from D.A., M.Shtrahman., and P.G. Surgeries were conducted by C.D. and M.Sehgal to provide experimental animals for testing. B.M. C.D. and L.Y. conducted experiments and analyzed data. L.Y. analyzed behavioral data and place field data in DG. All authors reviewed the manuscript.

## Competing interests

The authors declare no competing interests.
