## [Transparent Peer Review file · Nature Communications]

Open-source, high performance miniature 2-photon microscopy systems for freely behaving animals

Corresponding Author: Dr Peyman Golshani

Version 0:

Reviewer comments:

Reviewer #1

(Remarks to the Author)

This publication describes the development and manufacturing of a new 2-Photon miniscope that can be installed on the head of mice to obtain calcium signals from neurons located in deep brain areas such as the CA1 and the dentate gyrus in freely behaving animals. The advantage of this system is that it is open source and the files for building such a miniscope are available through the GitHub repositories. The weight of the system is 4 g, which is heavier than the 2P miniscope described in Zong et al., Cell 2023. Although the manufacturing of the system is well described, it lacks a more in depth investigations on proof of principles such as place cell characteristics (e.g. spatial tuning), lacking impact of the system on natural behavior (e.g. free-foraging, tortuosity during running). In summary, the scientific proof that defined neuronal properties such as place fields, spatial tuning, lack of changes in behavior and learning are underrepresented.

1. Despite the fact that it is great that such a miniscope is available to the field, the proofs that the weight of the system and the cable has no impact on the natural behaviour of mice came too short. How does the weight of 4 g which is higher than 3 g for the recently published 2P miniscope from Zong et al (Cell, 2023) influences natural behavior such as free-foraging, tortuosity during running, total distance run, time in the center of the home cage, just to give some examples for behavioral testing.

2. On the same lines, more detailed behavioral experiments are required to test for influences of the weight of the miniscope on natural behavior such as free-foraging, object recognition or spatial learning.

3. Comparing the CA1 video with the dentate gyrus video shows that different magnifications have been used. Please provide more detailed explanations on imaging settings.

4. The authors state that previous multi-photon imaging studies in the dentate gyrus of head fixed mice are limited due to lack of vestibular and other inputs that can modulate the activity of place cells and refer to Hainmueller & Bartos, Nature 2018. Although the reviewer agrees that in that study vestibular inputs are lacking due to head-fixation, the study showed that enrichment of the environment by tones, texture and tactile cues did not alter the characteristics of place cells or the number of place cells. Thus, the statement that 'this study did not examine whether other inputs can modulate neuronal activity' should be toned down. Moreover, the authors state on line 151 that they wished in their study to address these limitations ('this study did not examine whether other inputs can modulate neuronal activity') in their current study and refer to Figure 5. However, this figure shows exactly that has been previously shown, namely that granule cells have place fields. Thus, there is no apparent improvement presented in this compared to previous imaging studies in the dentate gyrus but also CA1. The respective sentence that 'limitations would have been addressed' in this study must be revised because that's not what this study by Madruga and colleagues did. On similar lines, it would be indeed important to show that the place cells in CA1 or any other hippocampal area show comparable spatial tuning as previously demonstrated and the same number of place cells or active cells as shown by other studies (single unit, 2-P head-fixed) in order to compare the outcome of the obtained data with previous work.

5. Please replace the wording multi-photon in line 148 with two-photon.

6. There is lacking evidence that the imaging was performed in the dentate gyrus. Indeed, the observation of apparently multiple place fields in Figure 5g suggests that other areas potentially the hilus (see Leutgeb et al., Science, 2007) have been imaged.

7. A craniotomy of 3.6 mm is quite large. Did such a large craniotomy have any impact on learning such as during object recognition, spatial memory or any form of environment/context related learning? See also above, point 2.

8. A large advantage of any form of 2-Photon microscopy in behaving mice is the ability to record the activity from the very same set of neurons across days. The removal and attachment of the miniscopes can be physically demanding to the animal and result in changes of the visual access, position of the visual field. How well and reliable can active neurons be identified on subsequent days? How stable is the system for daily attachment to and removal from the animals head?

Reviewer #2

(Remarks to the Author)

Madruza et al. present an open-source design for a miniature two photon microscope that weighs 4 grams and demonstrate its utility by recording calcium activity in the cortex, hippocampus, and dentate gyrus (DG). They show nice results on place cell analysis including video recording in the hippocampus, comparison of Ca activity in dendrites and somas in the retrosplenial cortex, and imaging at 650 microns depth below the coverslip to image the DG. However, the work is not novel given that there is already an open source design for the mini2P designed by Zong et al. (Zong, W. et al. Large-scale two-photon calcium imaging in freely moving mice. *Cell* 185, 1240-1256.e30 (2022)) and the technical developments have already been demonstrated in separate miniature microscopes. However, despite the lack of novelty, I do believe the work is important for the field of neuroscience, in particular it is important that different versions open-source designs for the microscope, hardware and software are available. The design incorporates different elements from the design of Zong et al., including an electro-wetting lens with significantly reduced cost from the piezo scanners in the Zong et al. design (although it cannot actuate as rapidly which should be mentioned). It also includes two silicon detectors on board without the need to deal with a fiber, although this was already demonstrated and published by the lab of Jason Kerr (Klioutchnikov, A. et al. A three-photon head-mounted microscope for imaging all layers of visual cortex in freely moving mice. *Nat Methods* 1–7 (2022) doi:10.1038/s41592-022-01688-9). Additionally, it is a benefit that the same miniature scope can be inserted into a cannula to reach hippocampus or DG while the Zong et al. mini2P requires purchasing a different objective, though this is not mentioned in the manuscript.

Unfortunately, I can not recommend the manuscript in its current form for publication in Nature Communications. The main reasons for this are due to the lack of technical novelty, the fact that this is not the first open-source miniature 2P microscope design available, and a lack of comparisons with other miniature multiphoton microscopes, missing technical details, and discussion of benefits/drawbacks.

My specific critiques are listed below:

1. The authors claim that their design is more robust to misalignment. They need to perform a tolerance model in Zemax to determine how robust it is to tilt or translation of the optics – either along or perpendicular to the optical axis, especially if they claim that it can be easily 3D printed.
2. The authors claim that their design can be easily reproduced in other labs, however off the shelf optics or even custom optics vary from their specs (for example, the diameters can vary by tens of microns) and may require iterations in the 3D print for a solid fit. For a design containing many optical elements and lenses, it will be very challenging for other labs to reproduce and would require many iterations in the 3D print. Additionally, see previous comment above – they should use Zemax to describe tolerance for each lens – the objective lenses in particular might require more strict alignment.
3. A comparison of the miniature recording with a benchtop 2P recording in vivo needs to be done to show that the same neurons and dendritic structures can be detected and whether there are image distortions. MEMS mirrors have distortions due to the coupling of the x and y axis. The authors do not mention how these were corrected or what driving function was used.
4. In order to drive the MEMS system at resonance, one requires the scanImage system and vDAQ at a cost of ~\$25-30k. In order to be upfront with the costs for other labs, the vDAQ expense should be included. Although the authors claim they will develop their own system, this has not been done. Additionally, the cost of the hollow-core fiber and dispersion compensation should also be included. The promise of being under \$10k is misleading.
5. The data in Fig. 4 in the retrosplenial cortex (RSC) taken over 30 minutes shows what appears to be some changes due to axial motion. Some of the dendrites in the initial image do not appear to be visible in the 16 min frame – maybe they are not firing but it is difficult to tell. The authors need to better quantify the axial motion, for example using a static fluorescent marker in the second color channel. They seem to have an additional mCherry marker, but do not use this. Additionally, there appears to be a regular striping pattern in the images in Figs. 4 and 5 which could make it challenging to do quantitative measurements. There is no explanation or discussion on what these are or why in the text.
6. The authors state that they use two on-board silicon-based photon detectors for “highly sensitive measurements”. However, this is not backed up with any evidence. The authors should perform a comparison of SiPM detection with multimode fiber coupled PMTs. SiPM detectors are prone to dark counts especially at room temperature and in previous work from the Kerr lab, time gated detection is used to mitigate this background signal. The authors should further elaborate on this.
7. Important parameters of NA, working distance, and axial scanning should be summarized up front, either in the abstract or in the introduction.
8. It would be helpful for many other labs if the authors compare the specifications of their miniature microscope with the device from Zong et al. *Cell* 2022. Perhaps including a table that lists the different parameters that could be in a supplement.

9. Minor: Since the authors only demonstrated two-photon microscopy, the title should indicate two-photon instead of multiphoton.

10. Minor: Since the authors already have measured the somatic and dendritic calcium transients, it would be interesting to compare the temporal profiles, for example using a spike template.

11. Minor: The manuscript describes a new fabrication method for the device, using SLA 3D Printer, but the details are not presented including the type of SLA 3D printer and resin.

Reviewer #3

(Remarks to the Author)

Reviewer #4

(Remarks to the Author)

The authors in the manuscript presented an open-source, high performance miniature multiphoton microscopy systems for freely behaving animals. The probe weighs approximately 4g and could record calcium dynamics from neurons located in deep structures and in dendrites over a $445\ \mu\text{m} \times 380\ \mu\text{m}$ FOV. Overall, the manuscript is methodologically robust and well organized to understand clearly with well-support engineering details. However, I think that by implementing a revision the authors can make this work better, and therefore I would like to recommend addressing the following:

(1) In the results section, at least three sets of imaging results for lateral and axial resolution measurement should be provided (these can be included in the Supplementary Information file); and imaging results for the field of view measurement should also be provided to assess the system's imaging uniformity and distortion.

(2) The manuscript uses HC-920 as the delivery fiber for femtosecond pulse, but the content on the characteristics of this fiber is relatively insufficient. The following details need to be further supplemented: structural parameters of the fiber, scanning electron microscopy (SEM), numerical aperture, transmission loss, bending loss, GVD parameters, etc.

(3) Information about the resonant frequency and scan angle of the MEMS scanner should be mentioned; the implementation logic of raster scanning (unidirectional/bidirectional) need to be further explained; and the resulting imaging frame rate should be introduced.

(4) It is recommended to compare the current work with recently reported miniature two-photon microscopy studies (Cell 185, 1240–1256 (2022); Nat. Methods 18, 46–49 (2021); Nat. Methods 14, 713–719 (2017)) to further highlight the advantages of this work.

(5) It is suggested that the authors further discuss the limitations of the current system metrics (such as imaging frame rate and imaging depth) and the next steps for optimization in the Future Directions section.

Version 1:

Reviewer comments:

Reviewer #1

(Remarks to the Author)

I am satisfied with the responses provided by the authors.

Reviewer #2

(Remarks to the Author)

The author's revisions have addressed most of my previous comments and the manuscript is greatly improved. However, there are a few minor points that still need to be addressed.

1. The authors twice mention in the manuscript that the head attached microscope will be under \$5k. However, this is complete conjecture. It is fine to mention (as they did) that the price of the most expensive component of the microscope (the \$4,000 objective) may come down with large scale orders. But stating that the price is under \$5K is incorrect and should be removed.

2. The authors highlight the fact that their device has two color detection, however there is cross-talk in the two channels because there are no emission filters in the design. To be transparent, the authors must report measurements of the cross-talk percentage and mention that unmixing is required for two color acquisition.

3. Supplemental table 2 is now included as a comparison between the different miniature 2P microscope designs. The authors should also include the weights of each design. Additionally, the authors report that their FOV is $445\ \mu\text{m} \times 300\ \mu\text{m}$, however in Table 1, they only report 300×300 as their maximum FOV for their acquisition. This should be corrected.

4. In the rebuttal, the authors reply:

"The reviewer also discusses a regular striping pattern present in some of the images, which is a regular and reproducible

artifact we have identified and now mention in the manuscript. This pattern is due, in our view, to capacitive coupling between the high voltage driving waveform (on the x-axis specifically) and the small amplitude SiPM signal currents that pass alongside the high voltage drive signal prior to transimpedance amplification."

I read through the revised manuscript carefully but did not see that this was addressed anywhere. The authors need to include this important point about the capacitive coupling and the striping pattern.

Reviewer #3

(Remarks to the Author)

Reviewer #4

(Remarks to the Author)

All questions raised in the previous round have been revised, and there are no additional questions this time.

Version 2:

Reviewer comments:

Reviewer #2

(Remarks to the Author)

The authors have addressed all of my concerns and I recommend the manuscript for publication in Nature Communications.

Response to Referees Letter (Madruga et al 2025, Nature Communications)

We are grateful that all three reviewers were excited about our work. Reviewer #1 was enthusiastic about the system design and the extent of the fabrication details within the manuscript. Reviewer #2 was quite complementary regarding the imaging results and the importance of the work to the field of neuroscience. Reviewers #3 and #4 stated that the device is methodologically robust and that the manuscript provided well-supported engineering details. Overall, we are grateful to all Reviewers, whose thoughtful comments have resulted in revisions that have greatly strengthened the manuscript. We now provide the results of extensive new experiments and analysis to address all of the reviewers' concerns. Specifically, we have performed new behavioral analysis to quantify the impact of the surgical preparation and 2P miniscope on free foraging behavior. We have also performed new experiments to show the ability of the 2P miniscope to record the same DG neurons over days. We also provide a new tolerancing analysis to show that our microscope is resilient to small alignment errors that may occur in batches of off-the-shelf optical components and 3D printed housings. We provide new experiments to quantify the increased fluorescent signal collection efficiency of the silicon photodetector strategy we take, compared to the more conventional fiber-array coupled PMT approach. Finally, we provide new tables outlining the cost of the microscope and all the associated equipment and software.

Below we provide a point-by-point response to each reviewer's critiques and mention how these critiques were addressed within the manuscript itself. All Reviewer comments will be highlighted in gray, and responses will be below.

Reviewer #1 (Remarks to the Author):

Reviewer 1 Summary:

Although the manufacturing of the system is well described, it lacks a more in depth investigations on proof of principles such as place cell characteristics (e.g. spatial tuning), lacking impact of the system on natural behavior (e.g. free-foraging, tortuosity during running). In summary, the scientific proof that defined neuronal properties such as place fields, spatial tuning, lack of changes in behavior and learning are underrepresented.

Reviewer 1 Comment 1:

1. Despite the fact that it is great that such a miniscope is available to the field, the proofs that the weight of the system and the cable has no impact on the natural behaviour of mice came too short. How does the weight of 4 g which is higher than 3 g for the recently published 2P miniscope from Zong et al (Cell, 2023) influences natural behavior such as free-foraging, tortuosity during running, total distance run, time in the center of the home cage, just to give some examples for behavioral testing. 2. On the same lines, more detailed behavioral experiments are required to test for influences of the weight of the miniscope on natural behavior such as free-foraging, object recognition or spatial learning.

We agree with the reviewer that increased weight of the microscope could impact behavior. We would like to note that the weight of our 2P miniaturized microscope is similar to the V3 UCLA Miniscope that was used by a large number of labs before the advent of the lighter V4 miniscope. In general, we believe that any head mounted device will to some extent alter natural behavior of the animal. As suggested by the reviewer we now conduct new experiments and thorough behavioral analysis to compare control animals (without the head mounted miniature microscope) to those being recorded with it. We now measure speed and total distance traveled in animals engaged in free foraging in an open field. Mice were freely able to navigate an approximately 38cm x 28cm chamber for the duration of the recording while sucrose pellets were supplied once every 120 seconds at a random location throughout the behavioral chamber. The position and locomotion of the mouse were recorded by a behavioral camera positioned roughly 50cm above the chamber recording at 30 frames per second (FPS).

As expected, we find no significant reduction in overall speed or cumulative distance traveled as a result of the head mounted device. The trends in reduction of speed and cumulative distance were less than what Zong et al., recorded in 5g 2P miniscopes but slightly greater than the 3g Mini2P described by the authors [1]. These results are now

presented in Supplemental Figure 1 and discussed in the results. Of note, mice were able to freely explore the entire arena, climb on walls, jump, and even escape our 8-inch-tall behavioral chamber via jumping while wearing our 2P miniaturized microscope.

Supplemental Figure 1: UCLA 2P Miniscope does not interfere with free-foraging behavior in mice. a) Three days representative trajectories of one mouse running in an approximately 38cm x 28cm open field, with 2 trials per day: UCLA 2P Miniscope with cable assembly, control (no head mounted device or cabling). b) Accumulated distance over 30 min of running. Lines, mean across 2 mice (each animal includes 3 days trajectories). Shaded region, SD at all time points. Color indicates experimental condition. Note the similarity between UCLA 2P Miniscope and control group. c) Cumulative speed distribution over 30 min of running. Each cure shows one trial. Color indicates experimental condition. d) Scatter plot showing total travel distance in each condition (30 min each). Horizontal lines indicate mean; Error bars indicate SD. Colored dots, individual trials. Conditions are compared using paired t-test, ns $p > 0.05$. e) Scatter plot showing median speed in each condition (30 min each). Horizontal lines indicate mean; Error bars indicate SD. Colored dots, individual trials. Conditions are compared using paired t-test, ns $p > 0.05$.

Reviewer 1 Comment 2:

2. Comparing the CA1 video with the dentate gyrus video shows that different magnifications have been used. Please provide more detailed explanations on imaging settings.

We now provide detailed information on the imaging settings for each of the imaging conditions and configurations in Supplemental Table 1. However, the magnifications used for the CA1 and DG recordings were identical, (440um x 380um). Because the physical dimensions of the granule cells are smaller and they're far more compact compared to pyramidal cells in CA1, the image field appears larger in DG, when in reality it is not. This can be also seen in figure 4a where CA1 and DG are visualized in depth. The imaging settings and magnification are maintained across this stack in depth and the difference in somatic dimensions can be seen.

Image Parameters For Datasets:

Figure	Brain Region	Frame Rate	FOV Size (Approximate)	Resolution (Pixels)	Resonant Axis Drive Frequency
Figure 3, Panel D	CA1	8.62 Hz	~ 210um x 210um	512 x 354	1650Hz
Figure 4, Panel B	RSC	7.92 Hz	~ 285um x 285um	512 x 350	1650Hz
Figure 5, Panel C	DG	8.62 Hz	~ 206um x 206um	512 x 354	1650Hz
Supplemental Figure 2	DG	8.62 Hz	~ 206um x 206um	512 x 354	1650Hz
Supplemental Figure 3	CA1	8.62 Hz	~ 308um x 308um	512 x 354	1650Hz

Supplemental Table 1: Imaging Parameters for all datasets. Bidirectional scanning used in all datasets.

Reviewer 1 Comment 3:

3. The authors state that previous multi-photon imaging studies in the dentate gyrus of head fixed mice are limited due to lack of vestibular and other inputs that can modulate the activity of place cells and refer to Hainmueller & Bartos, Nature 2018. Although the reviewer agrees that in that study vestibular inputs are lacking due to head-fixation, the study showed that enrichment of the environment by tones, texture and tactile cues did not alter the characteristics of place cells or the number of place cells. Thus, the statement that 'this study did not examine whether other inputs can modulate neuronal activity' should be toned down. Moreover, the authors state on line 151 that they wished in their study to address these limitations ('this study did not examine whether other inputs can modulate neuronal activity') in their current study and refer to Figure 5. However, this figure shows exactly that has been previously shown, namely that granule cells have place fields. Thus, there is no apparent improvement presented in this compared to previous imaging studies in the dentate gyrus but also CA1. The respective sentence that 'limitations would have been addressed' in this study must be revised because that's not what this study by Madruga and colleagues did. On similar lines, it would be indeed important to show that the place cells in CA1 or any other hippocampal area show comparable spatial tuning as previously demonstrated and the same number of place cells or active cells as shown by other studies (single unit, 2-P head-fixed) in order to compare the outcome of the obtained data with previous work.

We concur that the statement 'this study did not examine whether other inputs can modulate neuronal activity' was too strong and did not provide adequate context. We have modified the statement to reflect the fact that Hainmueller & Bartos, Nature 2018 did in fact provide environmental enrichments, and the inputs we were describing were vestibular in nature.

Additionally, we would generally like to be explicit in stating that the purpose and prose selected for presenting the imaging results from the UCLA 2P Miniscope in DG were in fact not to make bold scientific claims or present new findings regarding the activity of DG during free behavior. They were selected to replicate existing findings and to provide evidence that the microscope described is a research tool that is ready to be applied to new experimental conditions, such as resolving activity from granule cells in DG during spatial learning tasks, and foraging. With that said, we clearly show that the microscope is capable of recording activity in 2D environments in the same neurons over days, which is beyond what has been shown in the literature.

Reviewer 1 Comment 4:

4. Please replace the wording multi-photon in line 148 with two-photon.

We have now replaced multi-photon with two-photon in the manuscript.

Reviewer 1 Comment 5:

5. There is lacking evidence that the imaging was performed in the dentate gyrus. Indeed, the observation of apparently multiple place fields in Figure 5g suggests that other areas potentially the hilus (see Leutgeb et al., Science, 2007) have been imaged.

We now include Rebuttal Figure 1 which compares images from a commercial benchtop 2P microscope (fitted with a calibrated z-stage) showing the axial structure of the hippocampal formation being resolved alongside images from the 2P miniature microscope, along with depth-markers at each image plane. From this, alongside the stereotaxic coordinates and morphology of the region resolved, we are highly confident that we are resolving granule cells of the dentate gyrus. Additionally, because the hilus is a region sparsely populated by rather large neurons we are confident that we are resolving granule cells of the dentate gyrus as the regions we image contain extremely dense neuronal populations with small cell bodies. Although there are differences in the total depth between the CA1 layer and DG between animals (452um vs. 620um) the same anatomical features, layer structure, and neuron types/density are seen in both datasets. We have found variability from animal to animal in terms of total depth to DG and believe this is a clear description of the fact that we are using the UCLA 2P Miniscope to image DG *in-vivo* during free behavior.

Rebuttal Figure 1: Comparison between benchtop and UCLA 2P Miniscope recordings cross the axial direction of the hippocampal formation *in-vivo*. a) Benchtop 2P microscope images, with depths for each axial plane as measured on a calibrated z stage. b) Images from the UCLA 2P Miniscope lifted directly from Figure 5.

Reviewer 1 Comment 6:

6. A craniotomy of 3.6 mm is quite large. Did such a large craniotomy have any impact on learning such as during object recognition, spatial memory or any form of environment/context related learning? See also above, point 2.

Any craniotomy to image deep structures such as the hippocampus is an inherently invasive procedure which should be minimized in size so as to limit any potential cognitive impairment of the animal. Our microscope was designed to use 3mm diameter cover-glass windows, as this methodology is widely adopted in the head-fixed 2P microscopy field and has been shown to not impair learning in head-fixed hippocampus dependent tasks [2,3]. Such surgical

procedures require approximately a 3.2mm craniotomy (in order to install the titanium cannular window with approximately 100 μ m of clearance on each face). The front element of the 2P miniature microscope objective assembly presented here was engineered to fit inside of the cannular implant, to access not just CA1 but also deeper brain regions like DG over a large collection NA (for the lens size) while maintaining a small amount of mechanical freedom to enable slight translations of the microscope for identifying an ideal FOV relative to the window surface. As such, the craniotomy diameter needed to be increased by roughly 10 percent, which serves as a way to optically access relevant tissue with the miniature objective. To support the fact that no cognitive impairments have been observed with these implants, we provide Rebuttal Figure 2 which compares the learning curves of animals during a working memory task without an implanted window (only received a fiber optic implant over RSC, no aspiration, 0.5mm bur hole in skull), a 3.2mm craniotomy with cortical aspiration (generally adopted [2,3]) and those with a 3.6mm craniotomy with cortical aspiration (as is used in this work). Comparing these three groups over learning showed no statistical difference in learning in a delayed olfactory non-match-to-sample task [2,3].

Rebuttal Figure 2: Impact of cranial windows on learning and task performance. To assess the impact of the UCLA 2P Miniscope imaging implant on learning and behavior we compared three groups of animals learning a working memory dependent, delayed non-match to sample task described in [2] and [3]. Animals were either without cortical aspiration / window (green), with a standard 3.2mm craniotomy used routinely in head-fixed two photon microscopy studies of working memory dependent tasks (pink) or a 3.6mm craniotomy used in the UCLA 2P miniscope (blue). Among these groups there is no significant difference in learning curves.

Reviewer 1 Comment 7:

8. A large advantage of any form of 2-Photon microscopy in behaving mice is the ability to record the activity from the very same set of neurons across days. The removal and attachment of the miniscopes can be physically demanding to the animal and result in changes of the visual access, position of the visual field. How well and reliable can active neurons be identified on subsequent days? How stable is the system for daily attachment to and removal from the animals head?

To address this concern, we have conducted additional longitudinal experiments where the same image field was resolved over multiple days after removing and reinstalling the microscope system. We used the package CellReg [5] to detect overlapping neurons day-to-day and are confident in the capacity of the microscope to return to the same imaging field of view for longitudinal studies. The results of these experiments are summarized in the added Supplemental Figure 2. The physical attachment process of the 2P miniature microscope is accomplished in approximately 2 minutes following base-plating. It was critical for us to design a system that minimizes the stress

imposed on the animal as a result of the attachment process, and as such the microscope generally fits into place quickly and easily, only requiring tightening of two small set screws for full mechanical stability. Generally speaking, mice have incredibly wide visual fields, and as such we designed the microscope to limit restriction of the visual field of the animal. Main components such as the tunable lens, scanner assembly and collimator were placed caudal to the footprint of the baseplate, mitigating the impairment of the visual field. Critically, we tried to strike a balance between pushing hardware away from the visual field and mitigating the net torque placed on the baseplate itself by shifting the center of mass away from the baseplate. It is our view that we settled on a microscope design which considers both factors and attempts to minimize their negative impacts on natural behavior.

Supplemental Figure 2: UCLA 2P Miniscope enables multi-day tracking of the same cells in DG. a) The same FOV from DG across two days. Red dashed squares show the zoom-in areas (bottom) with red arrows highlighting the same features. Images are the mean of registered frames from Suite2P [4]. Day 2 image is calibrated using CellReg [5]. b) Firing fields for a representative cell (yellow circles labeled ROI 1 in panel a) detected across two days. (Top) Trajectories with spike locations; (Bottom) corresponding color-coded rate maps.

Reviewer #2 (Remarks to the Author):

Summary:

Madruga et al. present an open-source design for a miniature two photon microscope that weighs 4 grams and demonstrate its utility by recording calcium activity in the cortex, hippocampus, and dentate gyrus (DG). They show nice results on place cell analysis including video recording in the hippocampus, comparison of Ca activity in dendrites and somas in the retrosplenial cortex, and imaging at 650 microns depth below the coverslip to image the DG. However, the work is not novel given that there is already an open source design for the mini2P designed by Zong et al. (Zong, W. et al. Large-scale two-photon calcium imaging in freely moving mice. *Cell* 185, 1240-1256.e30 (2022)) and the technical developments have already been demonstrated in separate miniature microscopes. However, despite the lack of novelty, I do believe the work is important for the field of neuroscience, in particular it is important that different versions open-source designs for the microscope, hardware and software are available. The design incorporates different elements from the design of Zong et al., including an electrowetting lens with significantly reduced cost from the piezo scanners in the Zong et al. design (although it cannot actuate as rapidly which should be mentioned). It also includes two silicon detectors on board without the need to deal with a fiber, although this was already demonstrated and published by the lab of Jason Kerr (Klioutchnikov, A. et al. A three-photon head-mounted microscope for imaging all layers of visual cortex in freely moving mice. *Nat Methods* 1–7 (2022) doi:10.1038/s41592-022-01688-9). Additionally, it is a benefit that the same miniature scope can be inserted into a cannula to reach hippocampus or DG while the Zong et al. mini2P requires purchasing a different objective, though this is not mentioned in the manuscript.

Unfortunately, I can not recommend the manuscript in its current form for publication in Nature Communications. The main reasons for this are due to the lack of technical novelty, the fact that this is not the first open-source miniature 2P microscope design available, and a lack of comparisons with other miniature multiphoton microscopes, missing technical details, and discussion of benefits/drawbacks.

Reviewer 2 Comment 1:

1. The authors claim that their design is more robust to misalignment. They need to perform a tolerance model in Zemax to determine how robust it is to tilt or translation of the optics – either along or perpendicular to the optical axis, especially if they claim that it can be easily 3D printed.

To address the reviewer's concerns, we have completed a full tolerance analysis including tilt and translation of each optical element and summarized the results in Supplemental Figures 3 and 4. In general, we find that the objective lens set is the most sensitive to misalignment (specifically the third, high power lens element). However, because this 3-lens system is a part of a rigid mechanical assembly (aluminum housing) which is aligned and metrologically tested from the manufacturer (Optics Technology, New York, USA) this aspect of the tolerancing is constrained in a practical sense. The other free optical elements, such as the scan lenses and tube lenses are able to perform with diffraction limited performance within a tolerance of +/- 75 um translation and +/- 0.25 degrees angular. Because of the high resolution of modern SLA based 3D printers, we are confident that labs will be able to assemble and use these high-performance microscopes with a high degree of success. We recognize Reviewer 2's second comment regarding SLA printing parameters and will address those below:

Supplemental Figure 3: Tolerancing analysis, from the scanner forward without objective displacements. The analysis was conducted by shifting positions and tilt on each lens assembly apart from the objective set. a) PSF of the center of the field (top panel), at a scan angle corresponding to half of the FOV (middle panel), and at the edge of the FOV (bottom panel). Overlaid lines show 50 monte carlo simulation runs of the tolerancing analysis. b) RMS wavefront error over 50 monte carlo simulations. c) Tolerance Editor for these results, note that objective parameters are excluded.

Reviewer 2 Comment 2:

2. The authors claim that their design can be easily reproduced in other labs, however off the shelf optics or even custom optics vary from their specs (for example, the diameters can vary by tens of microns) and may require iterations in the 3D print for a solid fit. For a design containing many optical elements and lenses, it will be very challenging for other labs to reproduce and would require many iterations in the 3D print. Additionally, see previous comment above – they should use Zemax to describe tolerance for each lens – the objective lenses in particular might require more strict alignment.

Supplemental Figure 4: Tolerancing analysis from the scanner forward with objective displacements specifically to understand contributions from objective misalignment in combination with other optical tolerances. a) PSF of the center of the field (top panel), at a scan angle corresponding to half of the FOV (middle panel), and at the edge of the FOV (bottom panel). Overlaid lines show 50 monte carlo simulation runs of the tolerancing analysis. b) RMS wavefront error over 50 monte carlo simulations. c) Tolerance Editor for these results with objective displacement and tilt.

We are grateful for this comment, it is certainly true that off-the-shelf lenses vary in diameter and focal length within a specific tolerance range, especially between production runs. Ensuring the miniature microscope lens assembly described within the manuscript performs with the specifications listed is critical, with considerations given to manufacturing variability. Another consideration we made was in the fact that different 3D printers (specific model or even a specific printer of the same model) hold varying degrees of feature resolution and accuracy. Moreover, the orientation in which these components are fabricated shift the accuracy of such features. As such, the 3D-printed microscope housings were designed such that the lens bores are very slightly undersized and are easily reamed to an appropriate diameter with low-cost precision fabrication tools. These details are now included in the Mechanical System section of the manuscript, and we have included a link to the project's GitHub repository with new, additional information regarding the assembly of the microscope and the process of reaming the scan and tube lens bores (without specialized equipment or alignment jigs) for fitting the lenses optimally. We have also uploaded the .form files we used to print components, which include the part orientation, supports, and printer and material settings that worked the best in our hands, to minimize the variability in 3D printing process between users of the same printer model. Likely these settings can be seen as a starting point for users with other printer models. The microscope

housings also have been designed to include internal arrowhead features to identify when the lenses are correctly seated, and this is pointed out in the assembly guide on GitHub. While these lenses are intended to be press-fit, we also recommend adding a very small amount of NOA68 optical glue to the edge of these lens elements to ensure long lasting mechanical stability after repeated free behavior experiments. Due to the reasonably lenient tolerances (as analyzed and quantified above) we believe that users will be able to find success building these systems, despite variability in lens specifications from various manufacturers and 3D printing accuracy.

Reviewer 2 Comment 3:

3. A comparison of the miniature recording with a benchtop 2P recording in vivo needs to be done to show that the same neurons and dendritic structures can be detected and whether there are image distortions. MEMS mirrors have distortions due to the coupling of the x and y axis. The authors do not mention how these were corrected or what driving function was used.

To address the reviewer's concern, we now include Supplementary Figure 5 in which a FOV was measured between the miniature 2P microscope detailed in the manuscript and a benchtop commercial system (Scientifica Vivoscope 2P) to assess distortions and compare resolution. We find that in general the miniature 2P is able to resolve the same fine features like dendrites and axons as the commercial system. The most clear performance gap between the two came in the form of optical sectioning, since the commercial microscope supports a larger excitation numerical aperture (0.5) the extent of the optical sectioning is finer (approx. 5um) in comparison to the miniature 2P microscope (approx. 10um optical sectioning over a ~0.37 excitation NA). As a result, some neurons with the benchtop device look to be more "donut" shaped, whereas with the miniature 2P scope most neurons appear filled, since the top and bottom of the membrane is contained within the axial extent of the excitation PSF.

The microscope does use a coupled, bi-axial integrated scanner and as such is prone to distortions as mentioned. In our case, we simply drove the MEMS scanner with Scanimage, using a traditional raster-scan profile and tried to compensate for distortions at the edge of the field by using a relatively conservative spatial fill fraction of ~0.9. When compared to the benchtop recordings (with a traditional galvo-resonant scanning configuration), the distortions are not appreciable and seem to not impact image quality or accuracy. We have added specific verbiage to the manuscript (Control Software section) that detail these considerations and also make clear to the reader the drive functions that are used to control the miniature 2P Microscope.

Supplemental Figure 5: Comparison of benchtop and UCLA 2P Miniscope fields of view *in-vivo*. a) Benchtop microscope with an excitation NA of 0.5 imaging GCaMP-expressing neurons in CA1. b) Same FOV measured with the UCLA 2P Miniscope resolving the same neurons and projections

Reviewer 2 Comment 4:

4. In order to drive the MEMS system at resonance, one requires the scanImage system and vDAQ at a cost of ~\$25-30k. In order to be upfront with the costs for other labs, the vDAQ expense should be included. Although the authors claim they will develop their own system, this has not been done. Additionally, the cost of the hollow-core fiber and dispersion compensation should also be included. The promise of being under \$10k is misleading.

We agree that fully operating the miniature microscope requires additional hardware and software that is separate from the head mounted device. To make this point clearer to readers, we have now included and referenced a new Table 2 in the main manuscript (included here as well for review) which quantifies the cost of all the equipment used in the imaging system. Our intent was to minimize cost as much as possible, in both the equipment on the bench, and the head mounted microscope itself, to reduce barriers to entry for labs as much as possible. One example of that commitment to lowering cost is building the hardware around the free, open-source version of ScanImage (as opposed to more expensive premium versions that would easily solve some engineering difficulties, as will be discussed in Reviewer 2, Comment 6.) and MBF/ NI electronics that are common to the vast majority of benchtop 2P setups. We hoped that we reached a balance between a setup that could be fabricated from the ground-up in labs that do not do 2P microscopy for minimal cost, and one that could make use of existing lasers and NI or MBF electronics in 2P microscopy experienced labs to mitigate cost even further. Unfortunately, the largest barrier to many labs is the cost of the laser source, and we have included those costs in the new table as well. Our previously existing cost table (Table 1) focuses on the cost of components of the microscope headpiece, as it is most prone to damage as a result of extended use during free behavior. We have now made modifications within the text to ensure readers have an accurate understanding of costs not just of the head mounted device, but all supporting equipment as well.

Sub-Assembly	Manufacturer	Quantity	Cost	Subtotal
Microscope Headpiece	UCLA 2P Miniscope, Custom	1	\$7,244	\$7,244
Multi-photon Laser	Coherent, Axon 920 1W, TPC	1	\$63,500	\$63,500
DAQ Electronics and Computer	MBF Bioscience	1	\$31,500	\$31,500
Bench-Top Laser Optics	ThorLabs	1	\$3,000	\$3,000
Interface Electronics	Digilent, Hamamatsu, Custom	1	\$2,000	\$2,000
HC Optical Fiber	NKT Photonics	2	\$1,200	\$2,400
Control Software	ScanImage (Free)	1	\$0	\$0
				\$109,644

Table 2: Complete cost breakdown for the UCLA 2P Miniscope system and all associated components. These costs reflect purchase prices in the past, and may be slightly different currently. Total of all components together is in bold font and represents the approximate cost of setting up the miniature 2P microscope system with no existing equipment besides an optical table.

Reviewer 2 Comment 5:

5. The data in Fig. 4 in the retrosplenial cortex (RSC) taken over 30 minutes shows what appears to be some changes due to axial motion. Some of the dendrites in the initial image do not appear to be visible in the 16 min frame – maybe they are not firing but it is difficult to tell. The authors need to better quantify the axial motion, for example using a static fluorescent marker in the second color channel. They seem to have an additional mCherry marker, but do not use this. Additionally, there appears to be a regular striping pattern in the images in Figs. 4 and 5 which could make it challenging to do quantitative measurements. There is no explanation or discussion on what these are or why in the text.

Quantifying axial motion, especially of a 2P microscope that is operating during free behavior is a very important consideration for us. As the reviewer mentioned, we have access to a static reporter in the second color channel and we now have included additional analysis to monitor the stability of the fluorescence from the mCherry reporter over the course of the free behavior. We specifically selected 63 mCherry expressing ROIs contained throughout the FOV and monitored their mean fluorescence over the course of the recording following linear unmixing to remove any fluorescence bleed-through between static and dynamic channels. All signals were normalized in order to compare between ROIs throughout the recording. If there were substantial axial motion present through the recording that exceeded the axial PSF, we would expect many of these ROIs to change intensity in unison (as a result of leaving the focal plane), but this is not seen in the data. This has been displayed in Supplemental Figure 6 and as such we are confident that the dendrites mentioned by the reviewer were not present simply due to a lack of activity, not due to significant changes in axial motion. The reviewer also discusses a regular striping pattern present in some of the images, which is a regular and reproducible artifact we have identified and now mention in the manuscript. This pattern is due, in our view, to capacitive coupling between the high voltage driving waveform (on the x-axis specifically) and the small amplitude SiPM signal currents that pass alongside the high voltage drive signal prior to transimpedance amplification. We have now included a custom-made signal filter to selectively remove this high frequency noise and included it in a Fiji macro that has been uploaded to the GitHub directory as “Preprocess.imj.” This valuable information is now also reflected in the manuscript, and the revised preprocessing script is online and freely accessible. Future designs of the microscope include a common-mode choke to remove these frequencies in hardware as opposed to as an image processing step.

Supplemental Figure 6: Monitoring Stability of mCherry Fluorescence over 32 Minutes of Free Behavior. 63 mCherry-expressing ROIs were monitored for the duration of the full 32-minute free behavior RSC recording. Localized areas of mCherry expression were selected and monitored over the course of the recording.

Reviewer 2 Comment 6:

6. The authors state that they use two on-board silicon-based photon detectors for “highly sensitive measurements”. However, this is not backed up with any evidence. The authors should perform a comparison of SiPM detection with multimode fiber coupled PMTs. SiPM detectors are prone to dark counts especially at room temperature and in previous work from the Kerr lab, time gated detection is used to mitigate this background signal. The authors should further elaborate on this.

We now address this concern through experimental measurements that will be described here methodologically for completeness. In our initial experimental setup (unpublished data), we had used highly flexible optical fiber bundles from SCHOTT (AO-ERP Part No. IB1651350) with a 1.65mm core (1.45mm quality area) and 1.35m length. While these fiber bundles are extremely flexible and relatively small in diameter, they add considerable size and weight to the tether that couples the optical microscope to the head of the mouse. We initially used these bundles for early prototypes of the microscope, but moved away from them for this reason. Additionally, due to the cylindrical geometry of the individual fiber optic cores and their associated cladding, the transmission efficiency of the SCHOTT bundle is limited. To first test this, a 532nm (approximate GFP wavelength) laser was used along with two adjustable apertures positioned roughly 200mm apart. The apertures were set to a diameter less than 1.45mm (approx. 1mm) and the laser was coupled to the center of the fiber bundle such that 100% of the light was interfacing within the quality area. The power just immediately before the fiber bundle was measured using a ThorLabs power meter (part no. PM100D, S132 sensor) that was calibrated and set to the appropriate 532 nm wavelength. Correspondingly, optical power measurements immediately after the fiber bundle (with no kinks or short bend radii) were measured (without changing the alignment conditions) and we arrived at an average transmission efficiency of 30.96% (+/- 0.36%) over 6 measurements.

We also measured the power transmission through the rest of the collection optics for a conventional, benchtop PMT setup that we initially built for the UCLA 2P Miniscope. This includes a NIR laser blocking filter (Chroma ET750-SP, same as the 2P miniscope), a 1P dichroic mirror to split green and red channels (Chroma T550LPXR, same as the 2P miniature microscope) and an emission filter (Chroma, ET510/80M, 2P miniscope does not use an emission filter due to size, thickness and weight reasons). There are also two high NA collection lenses in the benchtop collection unit, each are 16mm focal length condenser lenses (ThorLabs, ACL25416U). The first is to roughly collimate the fluorescent signal from the bundle before the filter cube, the second is after the emission filter to compress all the signal onto the active area of the detector. The transmission power measurements were repeated using the 532nm laser, and this time the power was measured from immediately before the optical fiber bundle, and at the focal plane of the second condenser lens, where the GaAsP PMT was typically located. These measurements show that approximately 50% of the collected signal is lost through the benchtop collection optics, putting the efficiency of the complete setup at 15.77% +/- 0.288% over 6 measurements.

In order to compare results with the SiPM based detector, we repeated these measurements but placed the detector housing described in the manuscript immediately before the position of the fiber bundle. Optical power of the 532nm laser following the two adjustable apertures was measured again, immediately before entering the detector housing. The power at the position of the SiPM was measured as well, resulting in a transmission efficiency of 71.39% +/- 0.55% over 5 measurements. As a result, we are confident that the fluorescent power reaching the detector (either an SiPM or GaAsP PMT) is approximately 4.4x higher using the detection scheme on the head of the mouse, rather than relaying the fluorescence to benchtop detectors.

Reviewer 2 Comment 7:

7. Important parameters of NA, working distance, and axial scanning should be summarized up front, either in the abstract or in the introduction.

We now provide these parameters in the abstract as requested.

Reviewer 2 Comment 8:

8. It would be helpful for many other labs if the authors compare the specifications of their miniature microscope with the device from Zong et al. Cell 2022. Perhaps including a table that lists the different parameters that could be in a supplement.

We now include Supplemental Table 2 which outlines clear specifications and differences between the main multiphoton miniature microscope devices in the literature. This also compares other miniature microscopes in the field, such as the 3P device from Dr. Jason Kerr's lab, and older microscope designs from Dr. Weijian Zong. This table was modified from [6] and is replicated here with minor modifications, with the original corresponding author's written approval.

Reference	Klioutchnikov, A. et al. Nat Methods (2020)	Klioutchnikov, A. et al. Nat Methods (2023)	Zhao, C. et al. Nat Methods (2023)	Zhao, C. et al. Opt Express (2023)	Zong, W. et al. Cell (2022)	Qian, L. et al. bioRxiv (2024)	Madruca, B.A. et al. bioRxiv (2024)
Number of photons	3p	3p	3p	2p	2p	2p	2p
Emission NA, Excitation NA	0.9, 0.48-0.54	0.9, 0.48/0.58	0.65, 0.55	0.45, 0.25	0.45-0.5, 0.45-0.5	0.95, 0.26	0.6, 0.36
Lateral Resolution	0.8 μm	1.18 / 1.11 μm	0.97-1.21 μm	1.47 μm	1.15-1.24 μm	0.97-0.99 μm	0.98 μm
Axial Resolution	5.3 μm	13.8 / 10.1 μm	7.21-8.51 μm	24.64 μm	12.8-17.8 μm	80.4-90.2 μm	10.18 μm
FOV	160 \times 160 μm^2	300 \times 300 / 200 \times 200 μm^2	400 \times 400 μm^2	1000 \times 788 μm^2	420 \times 420 / 500 \times 500 μm^2	423 \times 439 μm^2	445 \times 380 μm^2
Frame Rate	27.78 Hz @ 120 \times 120 pixels	10.6 Hz @ 273 \times 280 pixels	15.93 / 8.35 Hz @ 128 \times 128 / 200 \times 200 pixels	9 Hz @ 600 \times 512 pixels	40 / 15 Hz @ 256 \times 256 pixels	9 Hz @ 512 \times 512 pixels	8.62 @ 512 \times 354 pixels, 7.92 @ 512 \times 350 pixels

Supplemental Table 2: Comparison of key optical specifications from selected multiphoton miniature microscope systems in the literature. The following table was lightly modified from [6] and used with the consent of the corresponding author.

Reviewer 2 Comment 9:

9. Minor: Since the authors only demonstrated two-photon microscopy, the title should indicate two-photon instead of multiphoton.

We have replaced “multi-photon” with “two-photon” in the title of the manuscript.

Reviewer 2 Comment 10:

10. Minor: Since the authors already have measured the somatic and dendritic calcium transients, it would be interesting to compare the temporal profiles, for example using a spike template.

We agree with this comment and have addressed it through a revision to Figure 4 that specifically compares somatic and dendritic calcium transients (panel f). Generally, somatic calcium events are longer lasting and temporally broad when compared to the faster kinetics of the dendritic calcium events. We add specific to this in the main manuscript as well. The updated figure and caption are presented here for convenience and consideration.

Figure 4: Dendritic imaging in cortex during free behavior. a) Schematic drawing of the experimental preparation used to record from RSC and the curvature of the cortical column that makes resolving long dendrites in the cortex possible. b) Time-course of still frames over the course of the 32-minute recording, highlighting the axial stability of the system and ability to reliably track single projections over substantial timeframes. Axial stability is assessed in more detail within Supplemental Figure 7. c) Calcium signals from somatic (blue) and dendritic ROIs (red) plotted separately over the first 30 minutes of the recording. d) Aligned and averaged calcium events from dendritic and somatic ROIs. e) Pseudo-colored image exported from Suite2P which colors ROIs by aspect ratio. This is superimposed on the mean of the entire recording. f) Overlay of mean somatic and dendritic calcium dynamics aligned by peak fluorescence. g) Animal trajectory over the 32-minute experiment.

Reviewer 2 Comment 11:

11. Minor: The manuscript describes a new fabrication method for the device, using SLA 3D Printer, but the details are not presented including the type of SLA 3D printer and resin.

We now provide these details within the “Mechanical Design” portion of the document. In addition, as mentioned previously, we have updated the GitHub repository to include the .form files we use for printing, that include printer settings, resin type, build orientation, and other parameters like supporting material placement.

Reviewer #3 (Remarks to the Author):

Reviewer #4 (Remarks to the Author):

Summary:

The authors in the manuscript presented an open-source, high performance miniature multiphoton microscopy systems for freely behaving animals. The probe weighs approximately 4g and could record calcium dynamics from neurons located in deep structures and in dendrites over a $445\ \mu\text{m} \times 380\ \mu\text{m}$ FOV. Overall, the manuscript is methodologically robust and well organized to understand clearly with well-support engineering details. However, I think that by implementing a revision the authors can make this work better, and therefore I would like to recommend addressing the following:

Reviewer 3,4 Comment 1:

(1) In the results section, at least three sets of imaging results for lateral and axial resolution measurement should be provided (these can be included in the Supplementary Information file); and imaging results for the field of view measurement should also be provided to assess the system's imaging uniformity and distortion.

We have completed a benchtop comparison under the helpful suggestion of Reviewer #2 (point 3) as Supplemental Figure 5 and have included those in this revision. We have also added an additional Supplementary Figure 7 that show three additional datasets to assess the imaging FOV used for determining the lateral and axial PSF (200 nm fluorescent beads), and an image of corn stem used to measure the dimensions of the FOV. Together these additional datasets will be helpful in assessing the system's imaging uniformity and distortions.

Supplemental Figure 7: Additional Calibration Images. Left panel shows individual 200nm beads as well as projections from single beads in the y,x and x,z planes. The images presented are mean projections across the 40μm image volume, taking care to not analyze aggregates of microspheres. Bottom image in the FOV is a single frame near the focal plane. Middle panel shows 2μm beads. The right panel shows an image of corn stem which was used to experimentally measure the FOV by translating one corner of a clear aspect of the sample from edge-to-edge with a calibrated motorized stage and recording the displacement.

Reviewer 3,4 Comment 2:

(2) The manuscript uses HC-920 as the delivery fiber for femtosecond pulse, but the content on the characteristics of this fiber is relatively insufficient. The following details need to be further supplemented: structural parameters of the fiber, scanning electron microscopy (SEM), numerical aperture, transmission loss, bending loss, GVD parameters, etc.

This is a commercial fiber produced by NKT photonics, and we have modified the manuscript to include key elements of the fiber parameters, like the NA (measured), transmission loss (provided by manufacturer), bending loss (provided by manufacturer), and GVD parameters at the wavelength we are using (provided by manufacturer). These can be found in section (Optical Hardware).

Reviewer 3,4 Comment 3:

(3) Information about the resonant frequency and scan angle of the MEMS scanner should be mentioned; the implementation logic of raster scanning (unidirectional/bidirectional) need to be further explained; and the resulting imaging frame rate should be introduced.

We believe this is related to Reviewer #1's point number 3, which prompted us to include Supplemental Table 1 which includes detailed imaging parameters, such as resonant frequency, bidirectional raster scanning, and framerate. Much of this information is now also referenced in the manuscript as well. Thank you for this helpful suggestion.

Reviewer 3,4 Comment 4:

(4) It is recommended to compare the current work with recently reported miniature two-photon microscopy studies (Cell 185, 1240–1256 (2022); Nat. Methods 18, 46–49 (2021); Nat. Methods 14, 713–719 (2017)) to further highlight the advantages of this work.

We agree that highlighting this work within the current state of the art literature would make the pros and cons of the current system clearer to the reader within the current landscape of miniature multiphoton microscopy. We believe that Supplemental Table 2, as was suggested by Reviewer #2, point number 8 will make the advantages of this work clearer and put things into a more clear global reference frame. It is referenced in the Future Directions.

Reviewer 3,4 Comment 5:

(5) It is suggested that the authors further discuss the limitations of the current system metrics (such as imaging frame rate and imaging depth) and the next steps for optimization in the Future Directions section.

Future directions section of the manuscript has been expanded to express the physical limitations of the current microscope, and possible next steps to overcome or extend the capabilities of miniature microscopes in the field. We believe this is a great addition to the manuscript and are grateful to the Reviewer.

References:

- [1] Zong, Weijian, et al. "Large-scale two-photon calcium imaging in freely moving mice." *Cell* 185.7 (2022): 1240-1256.
- [2] Taxidis, Jiannis, et al. "Differential emergence and stability of sensory and temporal representations in context-specific hippocampal sequences." *Neuron* 108.5 (2020): 984-998.
- [3] Dorian, Conor C., Jiannis Taxidis, and Peyman Golshani. "Non-spatial hippocampal behavioral timescale synaptic plasticity during working memory is gated by entorhinal inputs." *bioRxiv* (2024).
- [4] Pachitariu, Marius, et al. "Suite2p: beyond 10,000 neurons with standard two-photon microscopy." *BioRxiv* (2016): 061507.
- [5] Sheintuch, Liron, et al. "Tracking the same neurons across multiple days in Ca²⁺ imaging data." *Cell reports* 21.4 (2017): 1102-1115.
- [6] Qian, Long, et al. "High-throughput two-photon volumetric brain imaging in freely moving mice." *bioRxiv* (2024): 2024-10.

RESUBMISSION REBUTTAL (Madruga et. al, Nature Communications 2025)

1. The authors twice mention in the manuscript that the head attached microscope will be under \$5k. However, this is complete conjecture. It is fine to mention (as they did) that the price of the most expensive component of the microscope (the \$4,000 objective) may come down with large scale orders. But stating that the price is under \$5K is incorrect and should be removed.

We agree with the reviewer and have edited the manuscript accordingly. We now indicate that our entire microscope headpiece can be built for less than \$7.5k USD currently with low production numbers and the price can be further reduced if parts can be purchased in higher quantities, but do not specify a number.

2. The authors highlight the fact that their device has two color detection, however there is cross-talk in the two channels because there are no emission filters in the design. To be transparent, the authors must report measurements of the cross-talk percentage and mention that unmixing is required for two color acquisition.

We have now directly calculated the unmixing coefficient (0.318) and report this twice within the manuscript. In addition, we also now provide a description of why unmixing was required for multicolor imaging, for the reasons the Reviewer described. We now state: "To minimize weight, the detector assembly does not include discrete emission filters, instead we use linear unmixing (coefficient = 0.318) after image collection to achieve multicolor imaging while mitigating cross-talk."

3. Supplemental table 2 is now included as a comparison between the different miniature 2P microscope designs. The authors should also include the weights of each design. Additionally, the authors report that their FOV is 445 μ m x 300 μ m, however in Table 1, they only report 300x300 as their maximum FOV for their acquisition. This should be corrected.

Weights are now added to Supplemental Table 2, and a description of the reason for reporting the smaller FOV is now included in the manuscript: "The FOV was measured by displacing a 4 μ m bead (T14792, position #1) from one edge of the image field to the opposite edge using an electronically controlled linear stage with high precision and digital readout. Measurements were taken for each independent lateral axis. This was measured at 445 μ m x 380 μ m. During free behavior experiments, the FOV was reduced to ~ 308 μ m x 308 μ m for reliability. More image parameter details can be found in Supplemental Table 1."

4. In the rebuttal, the authors reply:

"The reviewer also discusses a regular striping pattern present in some of the images, which is a regular and reproducible artifact we have identified and now mention in the manuscript. This pattern is due, in our view, to capacitive coupling between the high voltage driving waveform (on the x-axis specifically) and the small amplitude SiPM signal currents that pass alongside the high voltage drive signal prior to transimpedance amplification." I read through the revised manuscript carefully but did not see that this was addressed anywhere. The authors need to include this important point about the capacitive coupling and the striping pattern.

We regret this omission. We now include this in the manuscript and describe two possible ways forward: 1. A signal processing image filter which is designed to selectively filter the artifacts, or 2. A description of the newest version of the microscope which has solved this problem entirely in hardware. We now include in the manuscript: "Through our measurements (and the data presented) we noticed a reproducible low-amplitude striping artifact that we believe to be due to capacitive coupling between the high voltage driving waveform (on the x-axis of the MEMS scanner specifically) and the small-amplitude SiPM signal currents that pass alongside the high voltage drive signal prior to transimpedance amplification. We have included a custom FFT signal filter and built a Fiji macro around it, to selectively remove this high frequency artifact from images. These resources have been uploaded to the GitHub directory under Guides, Image Filtering as "FFT_Filter.tif", and "Preprocess_Filter.ijm" respectively. Of note, recent design revisions of the microscope features increased coaxial shielding and thus does not exhibit this artifact."